# An actin filament branching surveillance system regulates cell cycle progression, cytokinesis and primary ciliogenesis

Muqing Cao [1,6] ✉, Xiaoxiao Zou [1,6], Chaoyi Li[1,6], Zaisheng Lin[1], Ni Wang[1], Zhongju Zou[2], Youqiong Ye [3], Joachim Seemann[4], Beth Levine[2,5], Zaiming Tang [1] ✉ & Qing Zhong [1] ✉

Dysfunction of cell cycle control and defects of primary ciliogenesis are two features of many cancers. Whether these events are interconnected and the driving mechanism coordinating them remains elusive. Here, we identify an actin filament branching surveillance system that alerts cells of actin branching insufficiency and regulates cell cycle progression, cytokinesis and primary ciliogenesis. We find that Oral-Facial-Digital syndrome 1 functions as a class II Nucleation promoting factor to promote Arp2/3 complex-mediated actin branching. Perturbation of actin branching promotes OFD1 degradation and inactivation via liquid-to-gel transition. Elimination of OFD1 or disruption of OFD1-Arp2/3 interaction drives proliferating, non-transformed cells into quiescence with ciliogenesis by an RB-dependent mechanism, while it leads oncogene-transformed/cancer cells to incomplete cytokinesis and irreversible mitotic catastrophe via actomyosin ring malformation. Inhibition of OFD1 leads to suppression of multiple cancer cell growth in mouse xenograft models. Thus, targeting OFD1-mediated actin filament branching surveillance system provides a direction for cancer therapy.

Centrosome, pericentriolar materials, and their associated cytoskeleton network are tightly associated with primary ciliogenesis, cell cycle checkpoint, and cell proliferation, however, the detailed mechanisms are largely unknown. Centrosomes are the main microtubule cytoskeleton organizing centers that control mitosis and ciliogenesis in most animal cells. The biogenesis of cilia, centrosomes, and centrosome-associated cytoskeleton are tightly regulated during the cell cycle. During $G_0$ and $G_1$ phase, the centrosome migrates to the cell surface where the mother centriole nucleates nine doublet microtubule bundles to assemble a cilium[1,2]. The centrosome duplicates in S phase and matures in $G_2$ phase[3]. Before mitotic entry, the cilium is disassembled, which liberates the centrosome from the plasma membrane to facilitate spindle assembly during M phase[4]. Defects of cilia disassembly have been reported to affect $G_1$/S transition through unknown mechanisms[5–9]. Ablation of cilia formation by disruption of intraflagellar transport (IFT) machinery rescues $G_1$/S cell cycle arrest caused by the cilium disassembly defect[5,7–9]. The retardation of ciliary resorption has been shown to delay cell cycle progression to S or M phase after cell cycle reentry and the absence of cilia is supposed to be a feature of vagarious cell proliferation[4]. Consistent with this, a growing number of studies have shown that assembly of cilia is often suppressed in human cancers, though the suppressive mechanisms

[1]Key Laboratory of Cell Differentiation and Apoptosis of Chinese Ministry of Education, Department of Pathophysiology, Shanghai Jiao Tong University School of Medicine (SJTU-SM), Shanghai, China. [2]Center for Autophagy Research, Department of Internal Medicine, University of Texas Southwestern Medical Center, Dallas, TX, USA. [3]Shanghai Institute of Immunology, Department of Immunology and Microbiology, Shanghai Jiao Tong University School of Medicine, Shanghai, China. [4]Department of Cell Biology, University of Texas Southwestern Medical Center, Dallas, TX, USA. [5]Howard Hughes Medical Institute, University of Texas Southwestern Medical Center, Dallas, TX, USA. [6]These authors contributed equally: Muqing Cao, Xiaoxiao Zou, Chaoyi Li. ✉e-mail: muqingcao@sjtu.edu.cn; zaimingtang2017@shsmu.edu.cn; qingzhong@shsmu.edu.cn

and the intrinsical molecular links between cell proliferation and ciliogenesis remain unknown[10–15]. Oral-Facial-Digital syndrome 1, OFD1, is a ciliopathy-associated protein located at centrioles and centriolar satellites[16,17]. The complete loss of OFD1 causes hyper-elongation of the centriole and defects of ciliogenesis[17]. Our previous studies found that autophagy selectively degraded OFD1 localized at centriolar satellites to promote ciliogenesis upon serum deprivation and depletion of OFD1 initiates ciliogenesis in MEFs and non-ciliated MCF7 cancer cells[16], while aberrant accumulation of OFD1 attenuates cilia formation in multiple cells via unknown mechanisms[16,18]. Considering the suppressive role of OFD1 in cilia formation, there is a possibility that OFD1 may function to inhibit ciliogenesis in tumors and promote cancer cell proliferation.

Deregulation of microtubule-related organelles or microtubule network may disrupt the fidelity of chromosome segregation in cell division, leading to the risk of aneuploid progeny[19,20]. To maintain genomic integrity, the fidelity of microtubule and microtubule-related organelles is surveilled by several pathways that arrest the cell cycle progression of cells with errors through the activation of cell cycle checkpoints, including the $G_1/S$ checkpoint (also known as the restriction or start checkpoint), the $G_2/M$ checkpoint and the metaphase-anaphase transition checkpoint (also known as the spindle checkpoint)[20,21]. Recently, emerging observations suggest the centrosome also functions as the actin organizing center and nucleates actin filaments via the WASH/Arp2/3 complex[22–26]. Functionally, centrosomal actin network is proposed to regulate cilia formation on basal bodies and microtubule nucleation at the spindle poles[24,25,27,28]. Up to date, little is known about the mechanism by which the centrosomal proteins coordinate the Arp2/3 complex to tune local actin nucleation. Further, whether F-actin network regulated by centrosomal proteins contributes to cell cycle control remains largely unknown.

In this study, we propose a model in which OFD1 functions as a class II nucleation promoting factor (NPF) to regulate centrosomal actin network, and to orchestrate cell cycle progression, cytokinesis, and primary ciliogenesis. OFD1 interacts with the Arp2/3 complex via its C-terminal acidic domain and promotes the Arp2/3 complex activity in the presence of class I NPFs. Elimination of OFD1 or disruption of the OFD1-Arp2/3 interaction compromises actin nucleation and drives non-transformed cells into reversible quiescence to prevent further mitotic defects by an actin filament branching surveillance checkpoint. However, oncogenically transformed cells, as well as most cancer cells, fail to activate the checkpoint and undergo mitotic cell death. Consistent with the important role of OFD1 to ensure the fidelity of mitosis, increased OFD1 expression is found to be associated with malignant transformation in multiple cancer types. Depletion of OFD1 by RNA interference largely attenuates tumor growth in culture cells and mouse xenograft models. These findings proposed a crucial role of OFD1 in local actin filament branching surveillance checkpoint by sensing branched actin polymerization around centrosomes, through which OFD1 regulates cell quiescence and cytokinesis, which play important roles in malignant transformation and cancer progression in a broad spectrum of human cancers. Targeting OFD1 and/or its mediated actin filament branching surveillance system may provide a direction for cancer therapy.

## Results

### OFD1 functions as a class II NPF to promote centrosomal F-actin branching synergistically with class I NPFs

Actin filaments form highly dynamical branched, isotropic, or bundled networks in response to the local signaling activity of assembly and disassembly[29–31]. Emerging evidence illustrates the importance of actin filament branching at centrosomes in primary ciliogenesis, microtubule organization, mitotic spindle formation, and chromosome congression[24,32–37]. Disrupting actin polymerization surrounding centrosomes likely removes roadblocks for membrane vesicles carrying ciliary proteins to dock at centrosomes, or releases actin-binding proteins to cilia, thereby promoting ciliogenesis robustly[32,35]. Coincidently, reducing the centriolar satellite-localized ciliopathy protein Oral-Facial-Digital syndrome 1 (OFD1) promotes ciliogenesis in mammalian cells[16,38], the mechanism of which is largely unknown. The promotion of ciliogenesis by both disruption of actin filament branching and OFD1 degradation prompted us to investigate the potential link between OFD1 and actin filament branching.

Actin filament branching is catalyzed by actin nucleation factors, actin-related protein 2/3 (Arp2/3) complex, and NPFs. The class I NPFs, including Neural Wiskott-Aldrich syndrome protein (N-WASP) family proteins, strongly promote Arp2/3-mediated actin polymerization. The class II NPFs, including the Cortactin (CTTN) family proteins, do not vigorously promote Arp2/3-mediated actin polymerization per se, but synergistically promote Arp2/3-mediated actin polymerization and branching in the presence of class I NPFs[39–43]. We tested the interaction between a seven-subunit complex of purified porcine Arp2/3 and purified recombinant human full-length OFD1-Flag protein (Supplementary Fig. 1a) in an in vitro pull-down assay. The purified Arp2/3 complex was pulled down by recombinant OFD1-Flag, but not by empty Flag beads (Fig. 1a), suggesting a direct interaction between OFD1 and the Arp2/3 complex. Further, we expressed seven subunits of the Arp2/3 complex individually, and tested their interaction with OFD1 in a co-immunoprecipitation assay. OFD1 binds to ARP2 strongly and several other subunits to a lesser extent (Fig. 1b). ARP2 binding to OFD1 appears to be specific, since another centriolar satellite protein PCM1 failed to bind to ARP2 in a co-immunoprecipitation assay (Supplementary Fig. 1b). In addition to ARP2, OFD1 also interacts with F-actin in a co-pelleting assay (Fig. 1c) and WASP-like proteins in a co-immunoprecipitation assay (Supplementary Fig. 1c). These results indicate that OFD1 interacts with actin filament branching protein complexes.

The Arp2/3 complex is intrinsically a weak nucleator of actin polymerization and its activity can be dramatically increased by class I NPFs, or class II NPFs that weakly activate Arp2/3 but have a strong synergistic effect with class I NPFs[44,45]. The physical interaction between OFD1 and Arp2/3 complex suggests a role for OFD1 in actin filament branching. We examined the effect of purified OFD1 on the branching nucleation of the Arp2/3 complex using an in vitro actin-polymerization assay with pyrene-labeled monomeric actin (G-actin). First, we tested if OFD1 directly influences the nucleation activity of Arp2/3 in pyrene-actin polymerization assay. Consistent with previous reports[46], GST-VCA, a functional domain of class I NPF N-WASP, stimulated actin polymerization in the presence of the purified Arp2/3 complex. OFD1 further activated the Arp2/3 complex in the presence of VCA in a dose-dependent manner, while OFD1 failed to stimulate Arp2/3-mediated actin polymerization without VCA (Fig. 1d). OFD1-showed a weak stimulation effect on Arp2/3-mediated actin polymerization, at a degree much weaker than what was observed with class I NPF (GST-VCA) but was similar to that of class II NFP Cortactin (CTTN), especially when the concentrations of the Arp2/3 complex were increased (Supplementary Fig. 1d). Both OFD1 and CTTN have no significant effects on Arp2/3-mediated actin polymerization in the absence of VCA, nor do they have a synergistic effect on each other (Supplementary Fig. 1e). These data indicate that OFD1 functions as a class II NPF rather than a class I NPF.

We further ask if OFD1 facilitates the recruitment of the Arp2/3 complex to the centrosomes and promotes the centrosomal actin branching in vivo. We transfected non-targeted siRNA or OFD1 siRNA into non-transformed hTERT-RPE1 cells (referred to as RPE1 cells henceforth). Endogenous OFD1 and ARP2 colocalize at the centrosomal region, however, compared to that of the control siRNA cells, the centrosomal localization of endogenous ARP2 was dramatically reduced in OFD1 siRNA depleted cells (Fig. 1e). We also transfected non-targeted siRNA or OFD1 siRNA into RPE1 cells with EGFP-OFD1 and

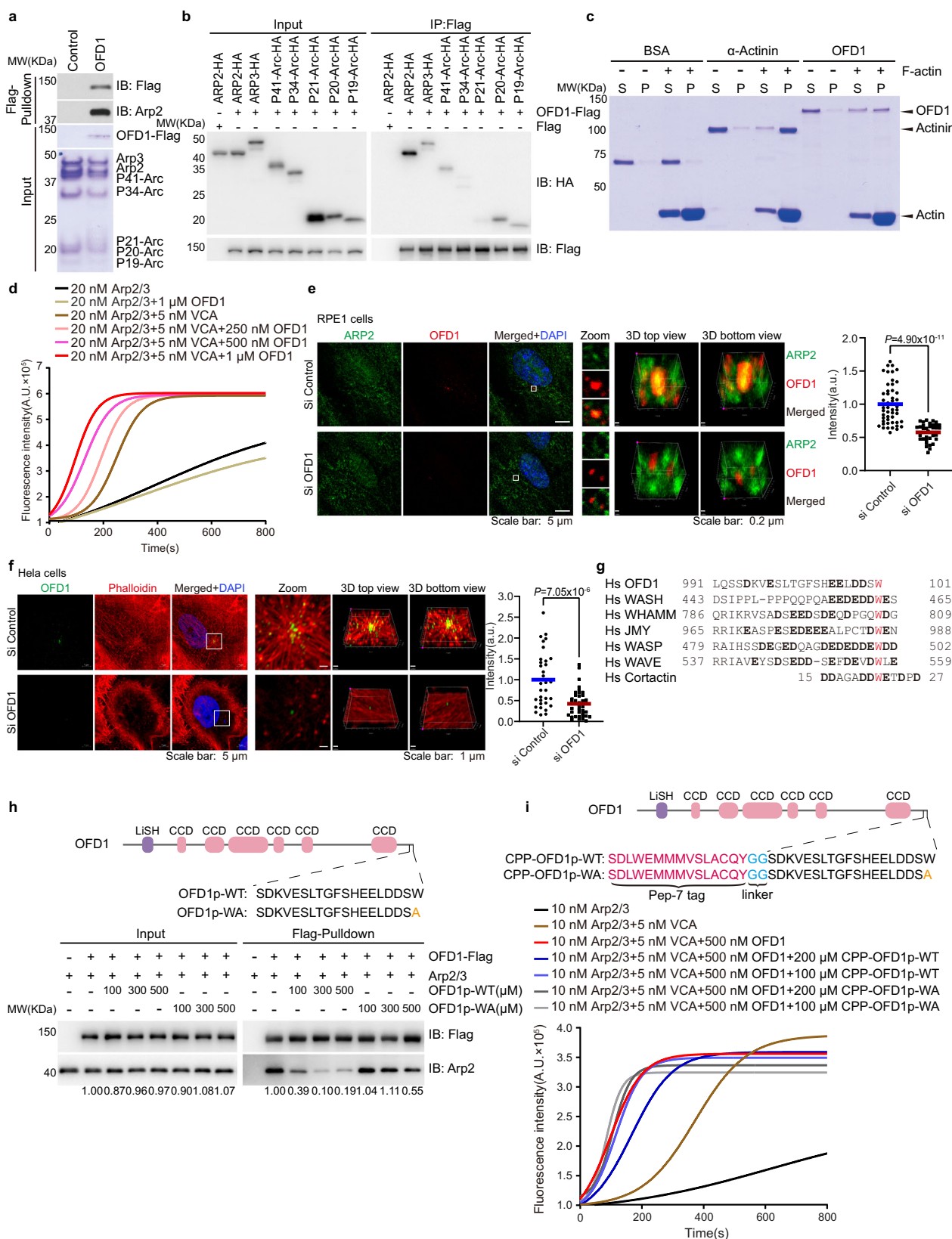

RFP-LifeAct, a fluorescence marker labeling F-actin in vivo. We found that centrosomes in cells with control siRNA were surrounded by a highly dynamic F-actin cloud, while loss of OFD1 largely reduced the F-actin cloud around centrosomes (Supplementary Fig. 1g and Supplementary Movies 1, 2), but had no obvious effect on the global distribution of F-actin networks (Supplementary Fig. 1f). HeLa cells have a

relatively low background of cortical actin, which makes them a better model system for observing centrosomal branching actin filaments. We also observed that the actin filaments surrounded centrosomes in HeLa cells, while the centrosomal actin filaments dramatically reduced upon OFD1 depletion (Fig. 1f). All these data indicate that OFD1 influences actin dynamics at centrosomes.

**Fig. 1 | OFD1 functions as a class II NPF to promote centrosomal actin branching. a** SDS-PAGE (Coomassie blue stained) and Immunoblot analysis of Flag pull-down samples, OFD1-Flag pulled down the purified 7-subunit Arp2/3 complex. **b** Immunoblot analysis of co-immunoprecipitation (IP) of OFD1-Flag with seven individual subunits of the Arp2/3 complex. **c** SDS-PAGE (Coomassie blue staining) analysis of F-actin pelleting assay. BSA and α-Actinin were used as negative and positive controls, respectively, S (Supernatant), P (Pellet). **d** OFD1 synergistic effect with class I NPF on actin polymerization. Polymerization of 0.875 μM 20% pyrene-labeled actin monomers was carried out in the presence of 20 nM Arp2/3 complex, 5 nM GST-VCA and 1000, 500 or 250 nM OFD1. **e** Representative imaging of ARP2 (green) and OFD1 (red) in RPE1 cells transiently transfected with control or OFD1 siRNA for 72 h and after fixation (Left Panel). 3D reconstruction of the zoom images (step size: 150 nm) were performed using Imaris Viewer software. ARP2 fluorescence integrated over a 1-μm-diameter circle around the centrosome for si-Control or si-OFD1 condition, 51 si-Control cells and 36 si-OFD1 cells examined over three independent experiments, $P = 4.9 \times 10^{-11}$, two-tailed unpaired student's t-test (Right Panel). **f** Representative imaging of endogenous OFD1 (green) and F-actin (phalloidin, red, Gamma-adjusted (0.5)) in HeLa cells transiently transfected with control or OFD1 siRNA for 72 h and after fixation with PFA-PEM (Left Panel). 3D reconstruction of the zoom images (step size: 140 nm) were performed using Imaris Viewer software. F-actin fluorescence integrated over a 3-μm-diameter circle around the centrosome for si-Control or si-OFD1 condition, 35 si-Control cells and 43 si-OFD1 cells examined over three independent experiments, $P = 7.05 \times 10^{-6}$, two-tailed unpaired student's t-test (Right Panel). **g** Sequence alignments of OFD1 and other NPFs. Conserved tryptophan residues are shown in red. **h** Peptide competition assay. Upper Panel, schematic representation of the structural domains of OFD1 peptides. Lower Panel, immunoblot analysis of pull-down assay of OFD1-Flag with the purified 7-subunit Arp2/3 complex (100 nM) with titration of OFD1p-WT peptides or OFD1p-W1012A (WA) peptides. The numbers under the gel lanes represent the ratio of ARP2 band intensity to Flag band intensity, which were normalized relative to the line 2 sample. **i** Actin polymerization upon OFD1 peptide treatment. Polymerization of 0.875 μM 20% pyrene-labeled actin monomers was carried out in the presence of 10 nM Arp2/3 complex, 5 nM GST-VCA, 500 nM OFD1 with 200 or 100 μM CPP-OFD1p-WT peptides, or with 200 or 100 μM CPP-OFD1p-W1012A(WA) peptides.

Next, we aimed to dissect the mechanism that underlying the OFD1-Arp2/3 interaction. Our bio-informatics analysis revealed that the C-terminus of OFD1 contains an acidic region characterized by a tryptophan (W) residue and several adjacent acidic amino acids, which shares remarkable similarity with the acidic domains of NPFs, including WASP-like proteins and Cortactin, that have been shown to be required for the interaction of these proteins with the Arp2/3 complex[40,47] (Fig. 1g). We further investigated whether the acidic domain of OFD1 mediates its interaction with the Arp2/3 complex and stimulates the Arp2/3 complex activity on actin polymerization. We generated two OFD1 variants bearing point mutations or truncation, one is OFD1 7 A mutant containing seven mutations (D996A, E998A, E1006A, E1007A, D1009A, D1010A, and W1012A) to replace all acidic residues and the tryptophan in this region with alanine, and the other is a truncated form of OFD1 (Δ950-1012) with the C terminal 62 amino acids of OFD1 deleted. Wild-type OFD1 co-immunoprecipitated with HA-tagged ARP2, while OFD1 7 A mutant or the truncation mutant dramatically reduced their interactions with ARP2 (Supplementary Fig. 1h). We then tested if the conserved tryptophan (W) residue in the acidic domain of OFD1 is crucial for the Arp2/3 complex interaction and activation. We substituted OFD1 W1012 residue with alanine, and found that this W1012A substitution diminished the interaction of OFD1 with ARP2 (Supplementary Fig. 1i). To further validate if the conserved tryptophan in OFD1 acidic domain is essential for centrosomal actin branching in vivo, we generated an inducible system in which RNAi-resistant EGFP-OFD1 and RNAi-resistant EGFP-OFD1 W1012A could be expressed in a Doxycycline dose-dependent manner. Notably, the expression of EGFP-OFD1 rescued OFD1 depletion-induced centrosomal actin debranching, however, the expression of EGFP-OFD1 W1012A failed to do so (Supplementary Fig. 1j). We synthesized a 19-amino acid peptide that derived from the acidic domain of OFD1 (19-amino acids at C-terminus of OFD1), which we named as OFD1p-WT. A tryptophan mutant peptide named OFD1p-W1012A was also generated. The presence of the OFD1p-WT peptide inhibited interactions between OFD1 and the Arp2/3 complex in a dosage-dependent manner in a pull-down assay, whereas the OFD1p-W1012A peptide presented much weaker inhibition on OFD1 and Arp2/3 interaction (Fig. 1h). The residual activity likely owes to partial binding of acidic residues. CPP-OFD1p-WT peptide, synthesized OFD1p-WT peptide in fusion to a previously characterized cell-penetrating peptide (CPP), inhibited the rate of OFD1-Arp2/3-VCA-mediated actin polymerization in a dosage-dependent manner, while CPP-OFD1p-W1012A peptide had a much weaker inhibitory effect on actin polymerization (Fig. 1i).

Both CPP-OFD1p-WT peptide and CPP-OFD1p-W1012A peptide showed no effect on basal actin polymerization mediated by the Arp2/3 complex only (Supplementary Fig. 1k). Interestingly, CPP-OFD1p-WT peptide, but not CPP-OFD1p-W1012A peptide, at least partially inhibited the rate of VCA catalyzed Arp2/3-mediated actin polymerization (Supplementary Fig. 1l). This may be due to the steric hindrance effect of CPP-OFD1p-WT peptide to GST-VCA upon binding to the Arp2/3 complex. Thus, the tryptophan in the acidic domain of OFD1 is crucial for its interaction with and stimulation of Arp2/3-VCA-mediated actin filament branching.

## OFD1 is dynamically regulated upon actin filament reorganization

OFD1 mainly localizes to centrioles and centriolar satellites[16,48]. We have previously shown that OFD1 protein at centriolar satellites is degraded upon serum starvation[16], a condition that actin polymerization undergoes dramatic reorganization[49–51]. We investigated if the stability of OFD1 at centriolar satellites would be affected by actin debranching. RPE1 cells were treated with cytochalasin D (Cyto D, inhibitor of actin filament organization), CK-666 (inhibitor of actin filament branching mediated by the Arp2/3 complex), CK-689 (inactive control of CK-666), and SMIFH2 (inhibitor of actin filament elongation mediated by formin), as well as serum starvation and siRNA against OFD1. Upon serum starvation, OFD1 was degraded from centriolar satellites as previously reported[16], similar to OFD1 siRNA treatment since OFD1 at centriolar satellites is short-lived[16] (Fig. 2a, Supplementary Fig. 2a). Both conditions led to primary ciliogenesis, which is consistent with our previous observation[16]. In CK-666-treated cells, OFD1 also disappeared from centriolar satellites and agglutinated into a few large centriolar aggregates along with cilia formation (Fig. 2a, Supplementary Fig. 2a), which has not been shown in CK-689-treated cells. This scenario was also observed in Cyto D-treated cells (Fig. 2a, Supplementary Fig. 2a).

To exclude the possibility that OFD1 aggregation is due to cell cycle oscillation, we synchronized cells by serum starvation 72 h before the treatment with actin inhibitors and analyzed the efficiency of synchronization by FACS. As shown in the figure (Supplementary Fig. 2c), cells remained efficiently synchronized after release and moved homogeneously through the cell cycle for 24 h. In synchronized cells, both endogenous OFD1 and PCM1, a well-established centriolar satellite marker, redistribute from discrete puncta to a few large centriolar aggregates under treatment with actin inhibitors (Supplementary Fig. 2d). The amount of OFD1 at centrosomes was quantified using the SUM z projection method on z-stacks of images in Image J (Supplementary Fig. 2d). The assembly of centrosome and the pericentriolar material are dynamically regulated and undergo liquid-liquid

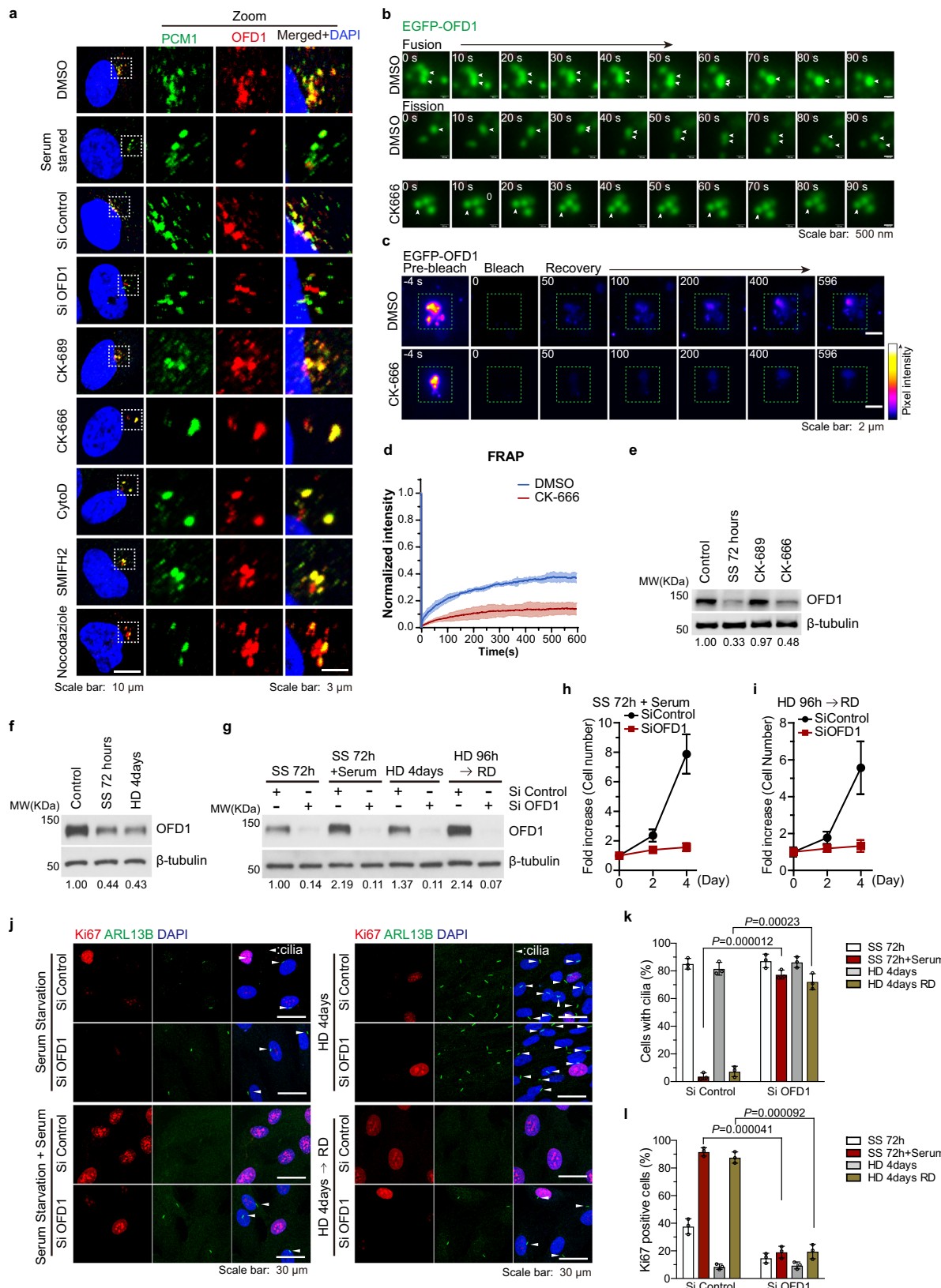

phase separation to allow many constituent molecules to diffuse and interact with one another efficiently[52–54]. In order to test whether OFD1 possesses the capacity to phase separate, we generated the Tet-inducible EGFP-OFD1-expressing RPE1 cells and measured the fusion and fission events in vivo. In DMSO treated cells, the EGFP-OFD1 droplets underwent rapid fusion and fission, while the EGFP-OFD1 droplets

remained stable upon CK-666 treatment (Fig. 2b, Supplementary Movies 3–5).

We further assessed EGFP-OFD1 dynamics by fluorescence recovery after photobleaching (FRAP) experiment. The FRAP results also indicated that there was less exchange of EGFP-OFD1 around the centrosome upon CK-666 treatment (Fig. 2c, d), suggesting a liquid-to-

**Fig. 2 | OFD1 is dynamically regulated upon actin filament reorganization.**
**a** hTERT-RPE1 cells were co-stained with antibodies against OFD1 (red) and PCM1 (green). Cells were subjected to DMSO, serum starvation, 120 μM CK-689, 120 μM CK-666, 100 nM Cyto D, 20 μM SMIFH2 treatment for 96 h, si-Control (control siRNA), si-OFD1 (siRNA targeted OFD1) treatment for 72 h or 15 ng/mL nocodazole treatment for 48 h. Scale bars: 10 μm, 3 μm (zoom in). **b** Live cell images of the fusion and fission events of centrosomal OFD1 in Tet-inducible EGFP-OFD1-expressing RPE1 cells treated with 0.1 ng/mL Doxycycline and DMSO or 120 μM CK-666 for 72 h. **c** FRAP analysis of EGFP-OFD1 condensates upon DMSO or CK-666 treatment (120 μM, 96 h) in Tet-inducible EGFP-OFD1-expressing RPE1 cells treated with 0.1 ng/mL Doxycycline, and the fluorescence recovery was recorded every 1 s for ~10 min. The dashed green squares indicate the bleached sites. **d** Plots of fluorescence intensity before and after photobleaching. Data shown represent mean ± SD, $n = 3$, and shaded areas show the standard deviation of the means. 50 cells examined over three independent experiments. **e** Immunoblot analysis of the protein levels of OFD1 and β-tubulin in hTERT-RPE1 cells that were subjected to DMSO, 120 μM CK-689, 120 μM CK-666 treatment for 96 h, or serum starvation (SS) for 72 h. The numbers under the gel lanes represent the ratio of OFD1 band intensity to β-tubulin band intensity, which were normalized relative to the line 1 sample. **f** Immunoblot analysis of OFD1 protein expression in exponentially growing RPE1 cells (control), 72 h serum-starved (SS) cells, and contact-inhibited cells induced by high density (HD) confluence. The numbers under the gel lanes represent the ratio of OFD1 band intensity to β-tubulin band intensity, which were normalized relative to the line 1 sample. **g** Immunoblot analysis of OFD1 protein

levels in RPE1 cells cultured within indicated conditions. Cells transfected with control siRNA or OFD1 siRNA were starved by serum deprivation for 72 h (Lane 1 and 2), or were plated at high density for 96 h (Lane 5 and 6). Starved cells were stimulated by serum addition for 30 h (Lane 3 and 4). Cells at high density for 96 h were passaged at a regular density for 30 h (Lane 7 and 8). The numbers under the gel lanes represent the ratio of OFD1 band intensity to β-tubulin band intensity, which were normalized relative to the line 1 sample. **h** Proliferation curves of serum-starved cells for 72 h, followed by 30 h of serum stimulation. Data shown represented as mean values ± SD, error bar was defined as SD. **i** Proliferation curves of cells cultured at high density for 96 h, and then passaged at regular density for 30 h. Data shown represented as mean values ± SD, error bar was defined as SD. **j** Representative images of immuno-staining for ARL13B (green) and Ki67 (red). RPE1 cells were transfected with siRNAs as indicated. In the Left Panel, cells were starved by serum deprivation for 72 h, followed by 30 h of 10% serum stimulation. In the Right Panel, cells were seeded at a high density to induce contact inhibition for 96 h, followed by passage at regular density for 30 h. **k** Quantification of ARL13B-positive cells described in (**j**). 300 cells examined over three independent experiments, $P = 0.000012$, $P = 0.00023$, two-tailed unpaired student's $t$-test. Data shown represented as mean values ± SD, error bar was defined as SD. **l** Quantification of Ki67-positive cells described in (**j**). 300 cells examined over three independent experiments, $P = 0.000041$; $P = 0.000092$, two-tailed unpaired student's $t$-test. All data shown represented as mean values ± SD, error bar was defined as SD.

gel transition. More strikingly, we found that, upon treatment of actin filament destabilizing agents CK-666 and Cyto D, not only OFD1, but also centriolar satellites proteins including PCM1 and BBS4, condensed into a few large puncta along with OFD1. This phenomenon was not observed in RPE1 cells treated with CK-689, serum starvation, SMIFH2, or Nocodazole (inhibitor of microtubule polymerization) (Fig. 2a, Supplementary Fig. 2a, b). Further, we asked whether OFD1 protein level is also reduced upon disruption of branched actin filaments. Inhibition of Arp2/3-mediated actin branching by CK-666, but not by its inactive control CK-689, leads to OFD1 reduction (Fig. 2e). This is also consistent with the results of immunofluorescence (Supplementary Fig. 2d). OFD1 reduction is likely mediated by degradation via both the autophagy-lysosome pathway and the ubiquitin-proteasome pathway as reported[16,55], since both CQ and MG132 treatment reversed the degradation of OFD1 by CK-666 (Supplementary Fig. 2f), suggesting that both the proteasome and lysosome pathways contribute to OFD1 degradation upon actin filament debranching. In contrast, the protein levels of centriolar satellite proteins BBS4 and PCM1 were largely unaffected (Supplementary Fig. 2f). RT-qPCR analysis showed that OFD1 mRNA was not decreased upon CK-666 treatment (Supplementary Fig. 2g). These data showed that when actin filament is debranched, part of OFD1 is degraded, and the rest undergoes a lipid-to-gel transition around centrosomes.

We further investigated the functional implication of OFD1 reduction. Both serum starvation and contact inhibition led to OFD1 reduction (Fig. 2f), and these treatments are known to cause cell cycle arrest and cilia formation. Supplementation of serum for cells under starvation or passage of cells in contact inhibition restored cell proliferation, promoted cilia disassembly, as well as recovered OFD1 protein levels (Fig. 2g–l). The restoration of cell proliferation and cilia disassembly upon serum supplementation is strictly dependent on OFD1 since cells depleted of OFD1 kept their cilia and failed to proliferate (Fig. 2g–l). The function of OFD1 in ciliogenesis is consistent with the previous report[16], and these results also suggest that OFD1 is likely a determining factor for cell cycle progression.

### Actin debranching phenocopies OFD1 depletion-induced quiescence in non-transformed cycling cells
We investigated the function of OFD1 in cell cycle progression by depleting OFD1 by siRNA in cycling RPE1 cells. Compared to control siRNA, treatment of siRNA targeting OFD1 led to robust inhibition of

cellular proliferation (Fig. 3a). Cell cycle analysis by flow cytometry showed that depletion of OFD1 caused cell cycle arrest before S phase (Fig. 3b). We performed the transcriptome analysis of OFD1-depleted RPE1 cells. RNA-seq data and immunoblots verified cell cycle arrest, characterized by the down-regulation of $G_1/S$ positive regulators and up-regulation of cell cycle inhibitors (Fig. 3c, Supplementary Fig 3a, b). Loss of *MKI67* transcription further suggested that OFD1-depleted cells exited the cell cycle and entered $G_0$ state (Fig. 3c), which was also supported by the loss of nuclear Ki67 antibody staining (Fig. 3d, f). OFD1 depletion also led to the formation of primary cilia, which is consistent with the previous report[16] (Fig. 3d, e).

In addition to siRNAs, we also generated a lentivirus-delivered Tet-On inducible knockdown RPE1 cell line, which initiates short hairpin RNA (shRNA) expression in response to Doxycycline (DOX). DOX treatment effectively depleted OFD1 protein expression in this inducible cell line. Consistent with the siRNA results, DOX-induced shRNA depletion of OFD1 (Fig. 3g), resulted in inhibited proliferation (Fig. 3h). We tested if the $G_0$ state caused by OFD1 depletion was reversible when the expression of OFD1 was restored. As cells were washed off DOX, they gradually restored OFD1 at centriolar satellites, disassembled their cilia, and regained the ability to proliferate (Fig. 3g–i). All these data demonstrate that, remarkably, loss of OFD1 is sufficient to drive cycling non-transformed cells into quiescence, even in the presence of rich nutrients. These data also indicate the effect of OFD1 depletion on cell cycle progression is reversible, therefore excluding the possibility that the OFD1-depleted cells undergo senescence or terminal differentiation.

Since OFD1 depletion leads to ciliogenesis[16], there is a possibility that OFD1 depletion-induced cell cycle arrest is due to primary ciliogenesis. Assembly and disassembly of primary cilium are strictly coordinated with the mitotic cell cycle. During interphase, cilium tightly fixes the mother centriole to the plasma membrane by distal appendages of the basal body. Before mitosis, cilium is disassembled, allowing disassociation of the centriole from the plasma membrane, which is expected to be essential for spindle formation[4]. Defects of cilia disassembly have been reported to affect $G_1/S$ transition through unknown mechanisms[5–9]. To determine whether the $G_0$ state induced by loss of OFD1 is cilia-dependent, we generated *IFT20*[-/-] RPE1 cells by CRISPR-Cas9 and further knocked down OFD1 in these cells. Deletion of *IFT20*, which encodes a member of the IFT-B complex, largely attenuated the formation of intact cilia, but Ki67 staining indicated

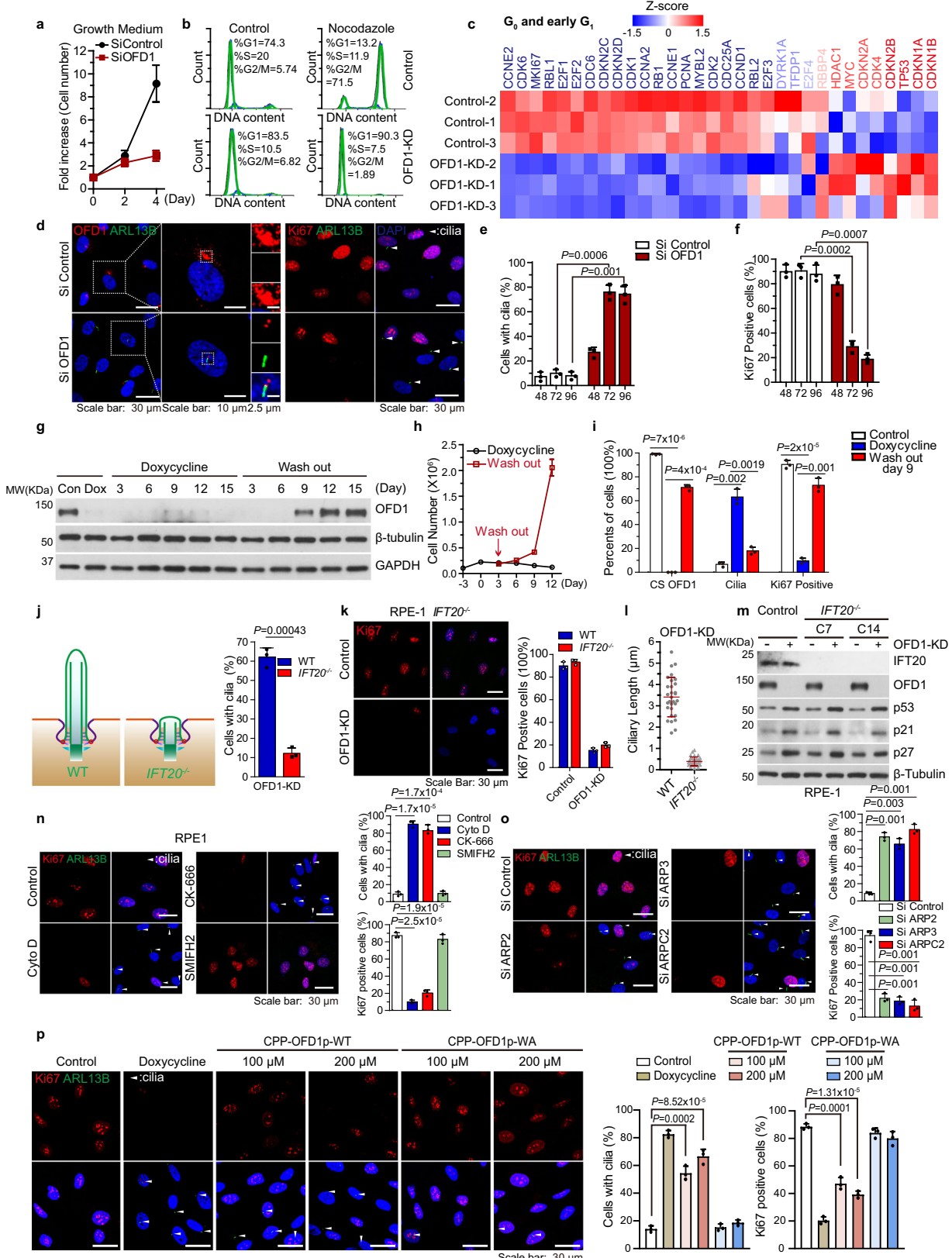

that loss of OFD1 still led to quiescence even without intact cilia (Fig. 3j–m). Lack of essential IFT components can not affect docking of the basal body to the plasma membrane or extension of ciliary transition zone[56], which means that the mother centriole can still be restricted to the plasma membrane upon OFD1 depletion. To assess if basal body docking is required for the quiescence caused by OFD1 loss,

we knocked out *CEP164*, encoding a distal appendage protein, which is essential for the docking of ciliary membrane vesicles at the top of the mother centriole[57,58]. Loss of *CEP164* blocked cilia formation (Supplementary Fig. 3c, d). CP110 was not removed from the mother centriole, indicating that the transition zone was not assembled (Supplementary Fig. 3e). Again, quiescence caused by OFD1 loss was not reversed in

**Fig. 3 | Actin filament debranching phenocopies OFD1 ablation in forcing non-transformed cycling cells into quiescence. a** Proliferation curves of RPE1 cells transfected with control or OFD1 siRNA in growing medium. **b** DNA content analysis by flow cytometry for RPE1 cells transfected with control or OFD1 siRNA, treated with 100 ng/mL nocodazole, or untreated. **c** Analysis of RNA expression of RPE1 cells and OFD1-depleted RPE1 cells. The transcription of genes related to $G_0$ and $G_1$ were shown in the chart. Each column is a biological replicate. The Z−score is shown by color key in the heatmaps. **d** Representative images of OFD1 (red) or Ki67 (red) with ARL13B (green) from RPE1 cells transfected with control and OFD1 siRNAs. Cilia are marked by arrowheads. **e** Quantified results of ARL13B for cilia formation in **d**. 300 cells examined over three independent experiments, $P = 0.0006$; $P = 0.001$, two-tailed unpaired student's t-test. **f** Quantified results of Ki67 for cell proliferation in **d**. 300 cells examined over three independent experiments, $P = 0.0002$; $P = 0.0007$, two-tailed unpaired student's t-test. **g** Tet-inducible OFD1 knockdown RPE1 cells. 200 ng/mL DOX was added into the medium to induce OFD1 shRNA expression at day 0, and then DOX was washed out with fresh medium twice at day 3. Immunoblot of OFD1, β-tubulin, and GAPDH protein levels in RPE1 cells with indicated treatment. **h** Proliferation curves of Tet-inducible OFD1 knockdown RPE1 cells by counting cell numbers. Data shown represented as mean values ± SD, error bar was defined as SD. **i** Data shown represent mean values ± SD percentage of cells with centriolar satellite pool of OFD1 or cilia or Ki67-positive staining. 300 cells examined over three independent experiments, $P = 7 \times 10^{-6}$; $P = 4 \times 10^{-4}$; $P = 0.002$, $P = 0.0019$; $P = 2 \times 10^{-5}$; $P = 0.001$, two-tailed unpaired student's t-test. **j, k** Knockdown of OFD1 by RNAi in $IFT20^{-/-}$ RPE1 cells decreased Ki67-positive cells with very less cilia formation. Data shown represent

mean ± SD percentage of cells with cilia or Ki67-positive staining. 300 cells examined over three independent experiments, $P = 0.0043$, two-tailed unpaired student's t-test. **l** The length of cilia from the indicated genotype of RPE1 cells upon OFD1 depletion. **m** Immunoblot analysis of indicated protein levels in WT or $IFT20^{-/-}$ RPE1 cell lines with or without OFD1 knockdown. C7 and C14 are different clones from two sgRNAs targeting $IFT20$. **n** Representative images of Ki67 (red) and ARL13B (green) staining in control RPE1 cells and cells treated with 120 nM Cyto D, 120 μM CK-666, or 15 μM SMIFH2 (Left Panel). Quantitative data for the staining (Right Panel). Data shown represent mean ± SD percentage of cells with Ki67-positive staining. 300 cells examined over three independent experiments, $P = 1.7 \times 10^{-5}$; $P = 1.7 \times 10^{-4}$; $P = 2.5 \times 10^{-5}$; $P = 1.9 \times 10^{-5}$, two-tailed unpaired student's t-test. **o** Representative images of Ki67 (red) and ARL13B (green) staining in RPE1 cells transfected with control, ARP2, ARP3, or ARPC2 siRNAs (Left Panel). Quantitative data for the staining (Right Panel). Data shown represent mean ± SD percentage of cells with Ki67-positive staining. 300 cells examined over three independent experiments, $P = 0.001$; $P = 0.003$; $P = 0.001$; $P = 0.001$; $P = 0.001$; $P = 0.001$, two-tailed unpaired student's t-test. **p** Representative images of Ki67 (red) and ARL13B (green) staining in Tet-inducible OFD1 knockdown RPE1 cells treated with 200 ng/mL Doxycycline, 100 or 200 μM CPP-OFD1p-WT peptides, or 100 or 200 μM CPP-OFD1p-W1012A(WA) peptides for 72 h (Left Panel). Quantitative data for the staining (Right Panel). Data shown represent mean ± SD percentage of cells with Ki67-positive staining. 300 cells examined over three independent experiments, $P = 0.0002$; $P = 8.52 \times 10^{-5}$; $P = 0.0001$; $P = 1.31 \times 10^{-5}$, two-tailed unpaired student's t-test. All data shown represented as mean values ± SD, error bar was defined as SD.

$CEP164^{-/-}$ RPE1 cells (Supplementary Fig. 3f, g). These data indicate that OFD1 loss-induced $G_0$ states in RPE1 cells are not caused by the formation of cilia or the association of centriole to the plasma membrane.

Disruption of the centrosome integrity blocks $G_1$-S progression by activating a centrosome surveillance checkpoint, which functions via the USP28-53BP1-p53-p21 signaling axis[59–62]. Loss of any member of this axis suppresses $G_1$ arrest upon centrosome damage. We asked if loss of OFD1 activates this signaling pathway leading to cell cycle arrest. Upon OFD1 loss, we found that centrosome numbers in cells depleted of OFD1 were not increased or decreased, indicating no strong centrosome duplication or loss occurred (Supplementary Fig. 3h, i). However, we observed that the p53-p21 pathway was activated (Supplementary Fig. 3j) and the transcription of several centrosome-related genes was downregulated (Supplementary Fig. 3a), suggesting that centrosome-associated signaling might be affected. We knocked out $TP53$, the gene encoding the major regulator of centrosome damage checkpoint protein p53, in RPE1 cells and treated these cells with centrinone, a PLK4 inhibitor that causes centrosome damage. Consistent with a previous study[59], centrinone induced centrosome loss caused cell cycle arrest in wild-type cells but not in $TP53^{-/-}$ cells (Supplementary Fig. 3h–j). On the contrary, deletion of p53 had a subtle effect on OFD1 loss-induced quiescence (Supplementary Fig. 3h, i), indicating that OFD1 loss activates a checkpoint pathway different from the centrosome surveillance checkpoint. Similar to that of OFD1 depletion, quiescence induced by the inhibition of F-actin dynamics (CytoD and CK-666 treatment, or ARP2 siRNA depletion) was not due to the presence of cilia or the centrosome surveillance pathway (Supplementary Fig. 3k, l). In summary, our data indicate that OFD1 loss activated cell cycle checkpoint is probably not due to cilia presence or a centrosome surveillance checkpoint.

We asked if the cell cycle progression controlled by OFD1 results from its regulation of actin filament branching dynamics. If this is true, we would expect inhibition of actin filament branching to phenotypically resemble OFD1 depletion. Indeed, inhibition of actin filament polymerization by Cyto D, and inhibition of actin filament branching by CK-666, but not inhibition of actin filament elongation by SMIFH2, strongly suppressed cell cycle progression and induced ciliogenesis (Fig. 3n). We also

synchronized cells by serum starvation 72 h before the treatment with actin inhibitors. And the results were similar to that of non-synchronized cells (Supplementary Fig. 2e). Depletion of three of the seven subunits of the Arp2/3 complex by RNAi caused the same phenotypes as OFD1 depletion (Fig. 3o). We introduced OFD1 peptides into RPE1 cells to determine if the OFD1-ARP2 interaction mediates the effect of cell cycle arrest. We observed that CPP-OFD1p-WT peptide, but not CPP-OFD1p-W1012A peptide, caused cell cycle arrest the same as depletion of OFD1 (Fig. 3p). These data demonstrate that OFD1 regulates cell cycle progression through actin filament branching mediated by OFD1 binding to and activation on the Arp2/3 complex.

## OFD1 ablation induced quiescence is reversed by oncogene activation and RB inactivation

To eliminate the concern of an off-target effect on siRNA, multiple siRNAs targeting OFD1 were used to deplete OFD1 from RPE1 cells. All of them led to cell cycle arrest accompanied by primary ciliogenesis, and the severity of these phenotypes correlated with OFD1 knockdown efficiency (Supplementary Fig. 4a–e). We aimed to complement the OFD1-depleted cells with ectopic OFD1 expression. As reported, overexpression of centriolar satellite proteins is prone to cause protein aggregation and dysfunction[63]. We designed an inducible system in which RNAi-resistant OFD1 and RNAi-resistant OFD1 W1012A could be expressed in a Doxycycline dose-dependent manner. When OFD1 was depleted by siRNA in these cells, both RNAi-resistant OFD1 and RNAi-resistant OFD1 W1012A expression were induced by titration of Doxycycline (Supplementary Fig. 4f). We found that only OFD1 expressing at appropriate levels restored the localization of OFD1 at centriolar satellites (Supplementary Fig. 4g), and rescued the defects of OFD1-depleted cells in ciliogenesis and cell cycle (Fig. 4a, b). OFD1 expression at lower or higher levels failed to do so (Fig. 4a, b). And as expected, OFD1 W1012A expression at tested levels failed to rescue OFD1 depletion-induced quiescence and ciliogenesis (Supplementary Fig. 4g, h). These data suggest a dynamic and precise regulation of OFD1 in ciliogenesis and cell cycle progression.

To determine whether the disruption of OFD1/F-actin-induced cell cycle arrest is a specific phenotype of hTERT-RPE1 cells or a general phenotype, we tested responses upon OFD1 loss or F-actin dynamics inhibition in several non-transformed cells, such as hTERT-BJ1 and

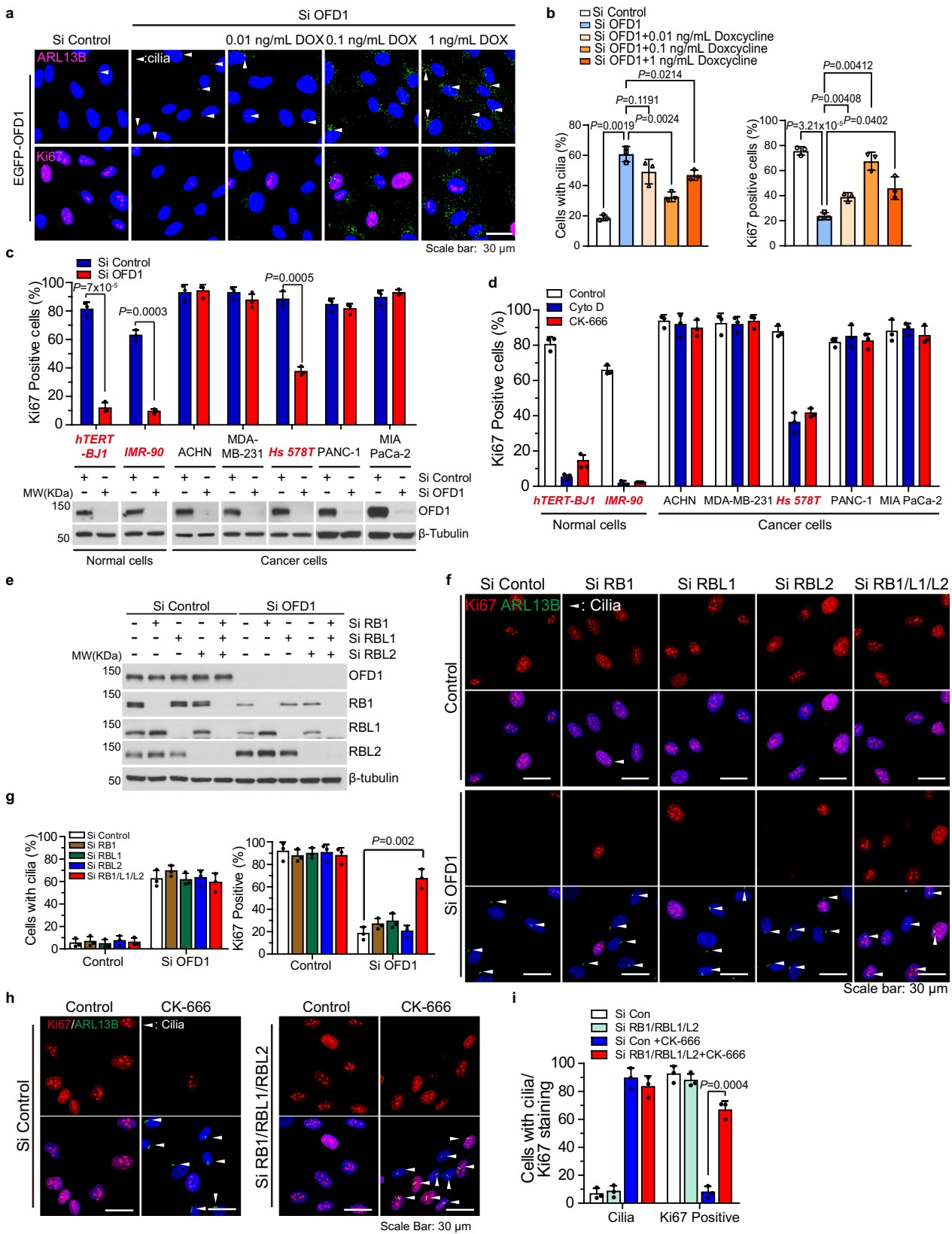

IMR-90, and in multiple types of cancer cells. Interestingly, only normal cells and Hs 578T showed significant cell cycle arrest, but not most cancer cell lines (Fig. 4c, d).

A major difference between normal cells and cancer cells is that cancer cells activate oncogenes to escape quiescence and retain the ability for persistent proliferation[64–66]. We investigated whether oncogene activation could bypass OFD1 depletion-induced cell cycle arrest. SV40 T antigen (TAg) is able to induce malignant transformation of normal cells by perturbation of the retinoblastoma (RB) and p53 tumor suppressor proteins[67,68]. We knocked down OFD1 in RPE1 cells stably transformed by TAg, and we observed that cilia were still formed, but $G_0$ arrest was abrogated in the presence of TAg

**Fig. 4 | OFD1 ablation-elicited cell cycle arrest is reversed by oncogene activation and RB inactivation in transformed cells. a** Representative images of Ki67 (magenta) or ARL13B (magenta) staining in Tet-inducible EGFP-OFD1-expressing RPE1 cells with indicated titration of Doxycycline and indicated siRNAs for 72 h. **b** Quantitation of cells positive for ARL13B or Ki67 staining in (**a**). Data shown represent mean values ± SD percentage of cells from triplicate samples. 300 cells examined over three independent experiments, $P = 0.0019$; $P = 0.1191$; $P = 0.0024$; $P = 0.0214$; $P = 3.21 \times 10^{-5}$; $P = 0.00408$; $P = 0.00412$; $P = 0.0402$, two-tailed unpaired student's *t*-test. Cilia are marked by arrowheads. **c** Normal or cancer cells were transfected with OFD1 siRNA and immunostained for Ki67 to mark proliferating cells. Cell lysates of RPE1 cells treated as indicated were immunoblotted for OFD1 and β-tubulin. Data shown represent mean ± SD percentage of cells positive for Ki67 are from triplicate samples. 300 cells examined over three independent experiments. **d** Normal cells and cancer cells were treated with 120 μM CK-666 or 120 nM Cyto D and immunostained for Ki67. Data shown represent percentage of cells positive for Ki67 are from triplicate samples. 300 cells examined over three

independent experiments. **e–g** RNAi knockdown of all three members of the RB protein family, RB1, RBL1, and RBL2, in human RPE1 cells abolishes OFD1 loss-induced cell cycle arrest but not cilia formation. **e** Immunoblot analysis of RPE1 cell lysates for the indicated proteins is shown. **f** Representative immunofluorescence staining of Ki67 (red) and ARL13B (green) in RPE1 cells with indicated treatments. Cilia are marked by arrowheads. **g** Quantitation of ARL13B or Ki67-positive cells in **f**. Data shown represent mean value ± SD, 300 cells examined over three independent experiments, $P = 0.002$, two-tailed unpaired student's *t*-test. **h** Representative images of Ki67 (red) and ARL13B (green) staining of RPE1 cells transfected with control siRNA or RB siRNA for 72 h. Cells were treated for 48 h with or without CK-666 24 h after siRNA transfection. Cilia are marked by arrowheads. **i** Quantitation of cells positive for ARL13B or Ki67 staining in **h**. Data shown represent mean ± SD percentage of cells from triplicate samples. 300 cells examined over three independent experiments, $P = 0.004$, two-tailed unpaired student's *t*-test. All data shown represented as mean values ± SD, error bar was defined as SD.

(Supplementary Fig. 4i, j). Since we showed that p53 is not responsible for OFD1 loss-induced $G_0$ arrest, this result suggests that TAg may bypass $G_0$ arrest by suppressing the RB pathway. This hypothesis was further supported by similar results observed when using the expression of another viral protein, E1A, which inactivates the RB pathway but not the p53 pathway (Supplementary Fig. 4k, l). To further confirm the function of the RB pathway in $G_0$ arrest induced by the disruption of OFD1/F-actin, we knocked down three RB family proteins, including RB1, RBL1 (also known as p107), and RBL2 (also known as p130), in non-transformed RPE1 cells. Knockdown of RB1 is insufficient for the OFD1-depleted cells to enter cell cycle, but knockdown of RBL1 and RBL2, along with RB1, abrogated the $G_0$ arrest upon OFD1 depletion (Fig. 4e–g). Similarly, to that of OFD1 loss, inhibition of F-actin dynamics by CK-666 failed to induce cell cycle arrest without RB proteins (Fig. 4h, i). These results suggest that depletion of OFD1 activates the actin branching surveillance system, which arrests cells in $G_0$ states through the RB pathway.

## Loss of OFD1 in transformed cells leads to cytokinesis failure and mitotic cell death

Oncogene-transformed cells are able to bypass cell cycle arrest caused by OFD1 loss, but whether this property leads to the growth advantage of cancer cells is unknown. We investigated the cell fate of SV40 Tag-transformed RPE1 cells (RPE1/TAg), which can enter mitosis without $G_0$ cell cycle arrest upon OFD1 depletion. To our surprise, the proliferation of OFD1-depleted, RPE1/TAg-transformed cells was suppressed rather than accelerated (Fig. 5a). The RPE1/TAg-transformed *CEP164*$^{-/-}$ cells also showed inhibited proliferation when OFD1 was depleted, indicating that this effect is not cilia-dependent (Fig. 5b). OFD1 depletion in RPE1/TAg-transformed cells did not cause centrosome loss (Supplementary Fig. 5a) and the proliferation inhibition of OFD1-depleted, RPE1/TAg-transformed cells was much more severe than that of centrinone-treated cells (Supplementary Fig. 5b).

We compared the DNA content of RPE1/TAg-transformed cells with or without OFD1 by flow cytometry and found that loss of OFD1 led to an increased proportion of tetraploid and octoploid cells (Fig. 5c). The increase of hyperploid cells indicates that the cells successfully replicated DNA in S phase, but failed to complete mitosis. We monitored mitosis of control and OFD1-depleted, RPE1/TAg-transformed cells by live cell imaging. Transformed RPE1/TAg cells rounded up, progressed through mitosis, and flattened out within 60 min (Fig. 5d, e, Supplementary Movie 6). By contrast, transformed RPE1/TAg cells with OFD1 depletion rounded up for a much longer duration before cytokinesis. A significant number of rounded cells elongated and formed the initial cleavage furrow, but failed to finish the cytokinesis step. These cells either flattened out and formed binucleate cells, or died during mitosis (Fig. 5d, e, Supplementary Movie 7). These results indicate that loss of OFD1 leads to mitotic defects in

transformed cells. We next labeled the cell skeleton by GFP-tubulin and RFP-LifeAct, which mark microtubule and F-actin networks in vivo, and observed mitosis by fluorescent live cell imaging. Upon the formation of spindles, control cells quickly assembled and contracted the actomyosin ring to finish cytokinesis (Fig. 5f, g, Supplementary Movie 8). Transformed RPE1/TAg cells with OFD1 depletion had no detectable defects in mitotic spindle assembly but displayed defects in actin accumulation at the onset of actomyosin ring formation, or at contraction of a formed actomyosin ring (Fig. 5f, g, Supplementary Movie 9), leading to cytokinesis failure, aneuploidy, and eventually cell death (Fig. 5h). We observed a significant portion of OFD1 translocating to the equatorial plate during anaphase, along with centriolar satellite marker PCM1 (Supplementary Fig. 5c, d), consistent with its role in mitosis. Taken together, OFD1-depleted, oncogenic-transformed cells bypass cell cycle arrest but fail to form functional actomyosin ring and undergo irreversible mitotic catastrophe, which leads to a preferential killing effect when OFD1 is inhibited in cancer cells.

## OFD1 sustains tumor cell proliferation and tumor growth

To test if OFD1 expression sustains cancer cell proliferation, we generated multiple DOX-dependent inducible OFD1 knockdown cancer cell lines. Upon DOX addition, the proliferation of most cancer cell lines was inhibited, correlating with the loss of OFD1 expression (Supplementary Fig. 6a, b). We have further validated the cell growth defects upon OFD1 inhibition in five cancer cell lines, including colon cancer (HT-29), lung cancer (A549), pancreatic cancer (PANC1), breast cancer (MDA-MB-231), and renal cancer (ACHN) (Fig. 6a–c). To further confirm that cancer cell growth inhibition is associated with the role of OFD1 in regulating actin dynamics, we treated PANC1 cells with CK-666 or an OFD1 peptide that disrupts the OFD1-ARP2 interaction. CK-666 has a similar effect on PANC1 cell growth inhibition compared to that of OFD1 depletion by shRNA; the CPP-OFD1p-WT peptide, but not the CPP-OFD1p-W1012A peptide, largely inhibited the proliferation of PANC1 cells (Fig. 6d). Since we observed that OFD1 depletion kills cancer cells but arrests normal cells in quiescence, we anticipated that the growth inhibition of OFD1 depletion should be reversible in normal cells but not in cancer cells. To verify this hypothesis, we performed a DOX washout experiment to temporally knockdown OFD1 in normal cells, RPE1, as well as multiple cancer cell lines, including ACHN, MDA-MB-231, and PANC-1. Strikingly, temporal targeting of OFD1 killed cancer cells irreversibly but normal cells grew back upon restoration of OFD1 expression (Fig. 6e).

Given the crucial role of OFD1 in cell cycle progression, we speculate that cancer cells increase OFD1 levels to maintain sustainable division. We analyzed OFD1 expression in cancers from the TCGA database. Compared with normal tissues, the expression of OFD1 aberrantly increased in multiple cancers, including colorectal cancer,

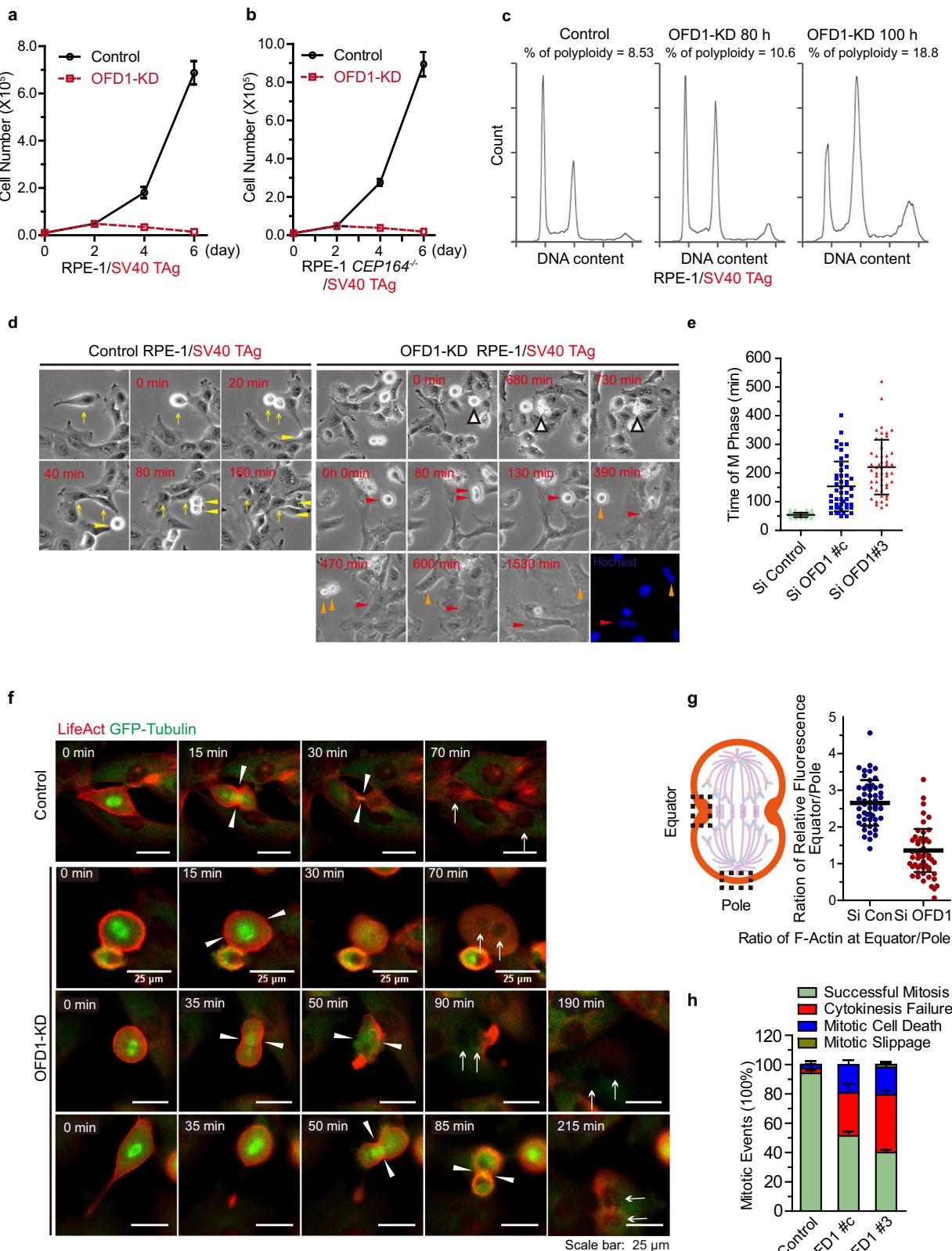

glioblastoma, renal cancer, liver cancer, lung cancer, and prostate cancer (Supplementary Fig. 6c). Consistent with the mRNA transcription pattern, OFD1 protein levels also increased in colon adenocarcinoma (COAD) and lung cancer (Supplementary Fig. 6d, e).

We further evaluated the effects of OFD1 depletion on tumorigenesis and progression in vivo, using mouse xenograft models with

highly malignant pancreatic cancer, colon cancer, and triple-negative breast cancer cells. Pancreatic cancer cell line PANC-1 with stably expressed Tet-inducible control shRNA or OFD1 shRNA were subcutaneously inoculated into NOD/SCID mice. DOX was delivered through their diet and drinking water to initiate shRNA expression. Xenografts from cells harboring OFD1 shRNA grew normally compared

**Fig. 5 | Loss of OFD1 in transformed cancer cells leads to cytokinesis failure and mitotic cell death. a** Proliferation curves of RPE1 cells expressing SV40 T Antigen with or without knockdown of OFD1. Data shown represented as mean values ± SD, error bar was defined as SD. **b** Proliferation curves of *CEP164−/−* RPE1/TAg cells with or without knockdown of OFD1. 300 cells examined over three independent experiments. Data shown represented as mean values ± SD, error bar was defined as SD. **c** Flow cytometry analysis of the DNA content of RPE1/TAg cells treated as indicated. Cells were stained with PI to show DNA content. **d** Mitotic events of WT (Left Panel) and OFD1-depleted (Right Panel) RPE1/TAg cells were monitored by phase-contrast time-lapse microscopy. Yellow arrows and arrowheads point at control cells undergoing normal M-phase. White arrows indicate OFD1-depleted cells undergoing mitotic cell death. Orange and red arrows point at OFD1-depleted cells with mitotic failure. Nuclei were labeled with Hoechst. **e** Quantitation of the

duration of M phase in cells transfected with indicated siRNAs. 50 cells examined over three independent experiments, two-tailed unpaired student's *t*-test. **f** Mitotic events of WT (Upper Panel) and OFD1-depleted (Lower Panel) RPE1/Tag cells were monitored by fluorescence time-lapse microscopy. RPF-LifeAct (red) and GFP-tubulin (green) were used to visualize F-actin and microtubules, respectively. Arrowheads point at the position of actomyosin rings. **g** Quantitation of the ratio of fluorescence signal of F-actin at the equator and pole in cells transfected with indicated siRNAs. Cells ($n \geq 50$) examined over three independent experiments, data shown represented as mean values ± SD, error bar was defined as SD, two-tailed unpaired student's *t*-test. **h** Quantitation of the indicated mitotic events of cells transfected with indicated siRNAs. Data shown represent mean ± SD. 300 cells examined over three independent experiments, data shown represented as mean values ± SD, error bar was defined as SD, two-tailed unpaired student's *t*-test.

to the xenografts from cells with control shRNA. With DOX treatment, the xenografts from PANC-1 cells expressing OFD1 shRNA grew at a much slower rate than xenografts without DOX treatment and control mice (no OFD1 shRNA expression) treated with DOX (Fig. 6f–h). We also evaluated the effect of OFD1 deficiency on the regression of established tumors. The DOX diet was started on day 24 post cancer cell implantation when the xenografts from cells with inducible OFD1 shRNA grew to an average volume of 200–300 mm³. Significant tumor regression was observed in mice with DOX-induced OFD1 knockdown, with tumors in six out of nine mice expressing OFD1 shRNA diminished and only three of nine mice expressing OFD1 shRNA developed measurable small-sized tumors, whereas all mice with the control diet developed tumors of much larger sizes (Fig. 6f–h). Similarly, to the pancreatic cancer xenografts, colon cancer xenografts from HT-29 cells (Fig. 6i–k) and triple-negative breast cancer xenografts (Fig. 6l–n) from MDA-MB-231 cells also relied on OFD1. OFD1 depletion by DOX largely attenuated cancer xenograft development in all three models. Moreover, extensive cell death was observed in the xenografts with OFD1 depletion 10 days post DOX addition (Supplementary Fig. 6f, g). Taken together, OFD1 is a promising therapeutic target for cancer treatment.

## Discussion

The eukaryotic cell cycle is controlled by a tightly regulated network to ensure that specific events take place to satisfy the needs of cell cycle progression. In this study, we propose that sufficient actin cytoskeleton branching is essential for cell cycle progression and the topology of the actin filament branching is monitored by an OFD1-dependent surveillance system. We report that OFD1 functions as a previously undescribed class II NPF, surveillants of actin cytoskeleton dynamics. Actin filament debranching caused by CK-666 or Cyto D treatment leads to OFD1 degradation and inactivation into liquid-to-gel-like structures, which also caused centriolar satellite proteins relocation. These results suggest that centrosomal actin network play a role in centriolar satellite maintenance or dynamics. Pericentrosomal preciliary compartment consists of membrane-less protein granules, multiple cytoskeleton network, and vesicles with diverse origins and serves as a hub to regulate key events around this region. The interplay between centriolar satellites and F-actin seems to be critical to the regulation and function of the pericentrosomal preciliary compartment. Loss of OFD1 or disrupting the OFD1-Arp2/3 interactions activate the cell cycle quiescence checkpoint in an RB-dependent manner in normal cells. Remarkably, the OFD1-mediated actin filament surveillance checkpoint is abolished upon oncogene activation. Most cancer cells are capable of escaping this checkpoint, probably due to the inactivation of the RB pathway. However, insufficient actin branching in cycling cancer cells ultimately leads to attenuated actomyosin ring formation, aborted cytokinesis, and cell death by mitotic catastrophe. OFD1 sustains cancer cell growth by promoting actin filament branching, and insufficient actin branching due to a lack of OFD1 decreases the growth advantage of cancer cells and results in

irreversible killing. Targeting OFD1 for destruction suppresses tumor cell growth in cultures and mouse xenograft models, indicating that OFD1 is an attractive target for cancer therapy.

The OFD1-mediated checkpoint is distinct from the centrosome surveillance checkpoint that is mediated by the USP28-53BP1-p53-p21 signaling axis[59–62], since no obvious centrosome defects were observed in OFD1-depleted cells. Although both checkpoints block $G_1$-S progression, the OFD1-mediated checkpoint is dependent on RB, while the centrosome surveillance checkpoint is reliant on p53. The precise mechanistic details of this OFD1-mediated checkpoint require further investigation.

During mitosis, remodeling of the actin network governs changes to cell morphology to fit cell cycle progression. OFD1 specifically localizes around the centrosomal region and the equatorial plate. The loss of OFD1 leads to defects in cytokinesis and actomyosin ring formation, likely through the Arp2/3 complex. Coincidently, both chemical and genome-wide genetic screen to identify cytokinesis targets and midbody proteomic analysis have found that the Arp2/3 complex were involved in cytokinesis[69]. The loss of OFD1 may function through the Arp2/3 complex to affect the stability of the actin branching network[70], or suppress excessive formin activity to inhibit the assembly of the actomyosin ring[71]. Another evidence for a role of OFD1 in mitosis comes from a previous report in which knockdown of OFD1 was found to cause mitotic delay and a binuclear/polylobed phenotype in a genome-wide phenotypic profiling of cell division[72].

Cancer therapy demands high efficacy and while also minimizing unwanted side effects. Our study using shRNA specific for OFD1 in the treatment of established tumors suggests that OFD1 loss could robustly inhibit tumor growth of pancreatic cancer cells. Most encouragingly, loss of OFD1 mainly kills cancer cells but only causes reversible cell cycle arrest in normal cells, suggesting that fewer side effects may occur upon the treatment. The actin cytoskeleton is an important target in the treatment of cancer, but chemotherapeutic targeting attempts have been largely hampered by high toxicity because of the general function of actin in cell homeostasis. The compartmentalization of OFD1 at the centrosomal or equatorial plate region may provide a specific chance to target the local actin cytoskeleton with limited effects in other regions, which may largely reduce the toxicity previously associated with targeting global actin network. Further studies of OFD1-Arp2/3 interactions, with the assistance of high-resolution structures, may help to design potential drugs in the future.

## Methods
### Chemical handling
The chemicals were used at the indicated concentrations: 100 ng/mL nocodazole, 200 ng/mL Doxycycline, 120 nM Cyto D, 120 μM CK-666, 120 μM CK-689, and 15 μM SMIFH2. Small compounds were dissolved in DMSO (Sigma-Aldrich, 276855). TAP buffer: 20 mM Tris HCl (pH 7.5), 150 mM NaCl, 0.5% Nonidet P-40, 1 mM NaF, 1 mM $Na_3VO_4$, 1 mM EDTA, protease inhibitor mixture (Roche).

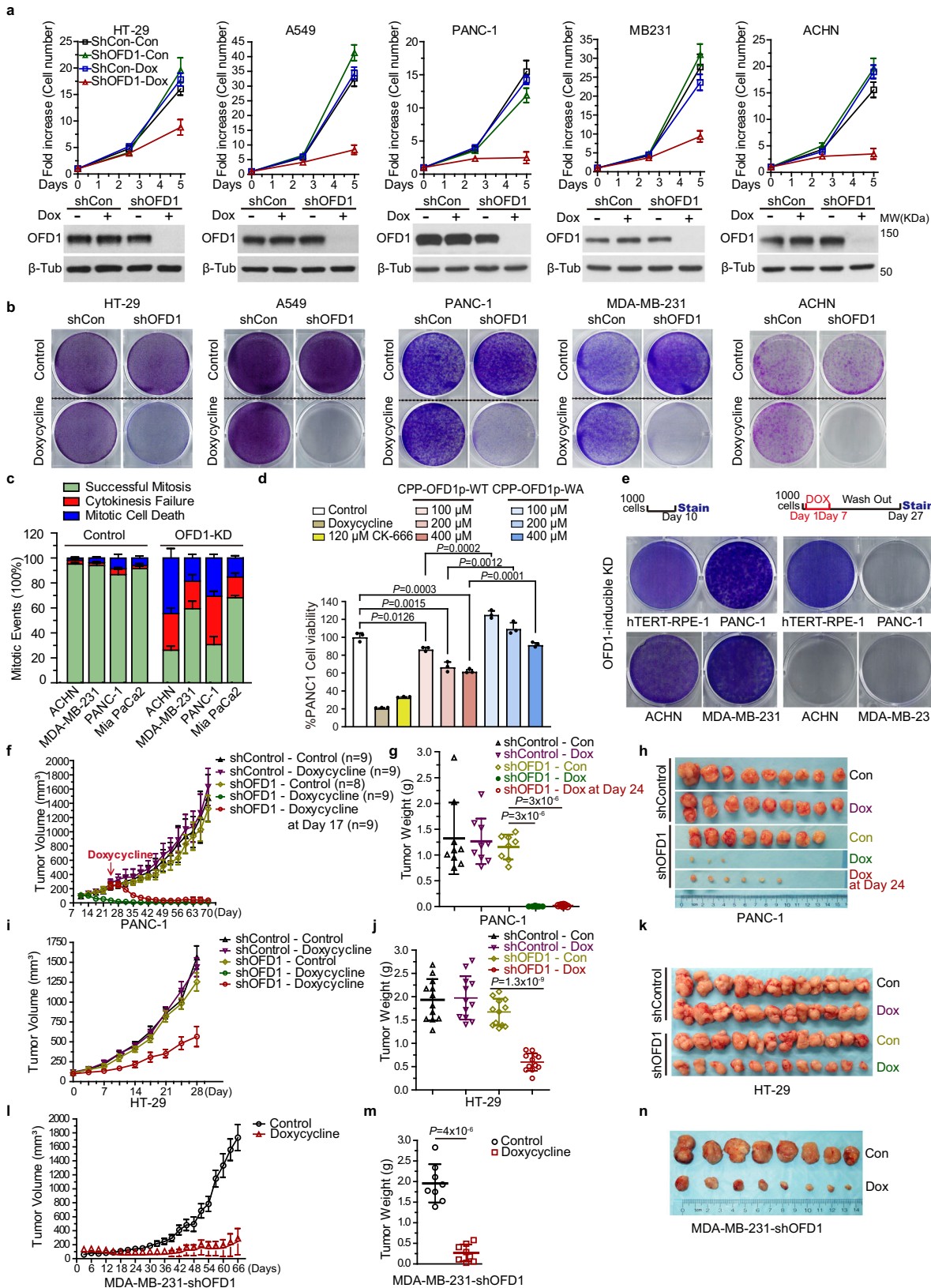

## Cell culture

hTERT-RPE1, ACHN, OCM-1, OCM-1a, OM431, and MDA-MB-175-VII cells were cultured in DMEM/F12 (Sigma-Aldrich, D8437) supplemented with 10% FBS (ExCell Bio, FSP500) and 100 U/ml Penicillin-Streptomycin (GIBCO, 15140122). HeLa, HEK293, hTERT-BJ1, IMR-90, Hs 578 T, MCF7, MDA-MB-231, MDA-MB-468, T47D,

HT-29, A549, Hs 766 T, MIA PaCa2, PANC-1, and PL45 cells were cultured in DMEM (Sigma-Aldrich, D6429) supplemented with 10% FBS and 100U/ml Penicillin-Streptomycin. HCC1937, HCC1143, HCC38, 769-P, and BxPC-3 cells were cultured in RPMI-1640 (Sigma-Aldrich, R8758) supplemented with 10% FBS and 100 U/ml Penicillin-Streptomycin.

**Fig. 6 | OFD1 sustains tumor cell proliferation and tumor growth. a** Proliferation (Upper Panel) and immunoblot (Lower Panel) analyses of indicated cell lines expressing Doxycycline-inducible control shRNA or OFD1 shRNA. Data shown represented as mean values mean ± SD, error bar was defined as SD. **b** Proliferation assay of indicated cell lines expressing control shRNA or OFD1 shRNA. Cells were stained with crystal violet. **c** Quantitation of mitotic events in the indicated cell lines with or without knockdown of OFD1. 300 cells examined over three independent experiments, data shown represented as mean values ± SD, error bar was defined as SD, two-tailed unpaired student's *t*-test. **d** Quantification of the percentages of cell viability upon OFD1 peptide treatment as indicated for 7 days. 300 cells examined over three independent experiments, $P = 0.0126$; $P = 0.0015$; $P = 0.0003$; $P = 0.0002$; $P = 0.0012$; $P = 0.0001$, two-tailed unpaired student's *t*-test. Data shown represented as mean values mean ± SD, error bar was defined as SD. **e** Proliferation assay of indicated cell lines expressing OFD1 shRNA. Left Panel: Cells were grown in the absence of Doxycycline for 10 days. Cells were grown in the presence of Doxycycline for 6 days, DOX was then removed and the cells grown for 20 days. Cells were visualized by crystal violet staining. **f** Tumor volume of xenografts formed after subcutaneous injection of NOD/SCID mice with PANC-1 cells expressing control shRNA or OFD1 shRNA in the presence or absence of Doxycycline in the animal diet. Tumor size was measured twice per week. 9 mice per group per time point examined over three independent experiments, two-tailed unpaired student's *t*-test. Data shown represented as mean values ± SD, error bar was defined as SD. **g** Weight of PANC-1 xenograft tumors from experiments shown in **f**. Data shown represented as mean values ± SD, error bar was defined as SD, $P = 3 \times 10^{-6}$; $P = 3 \times 10^{-6}$, two-tailed unpaired student's *t*-test. **h** Tumor images of indicated PANC-1 xenograft tumor genotype from experiments shown in (**f**). **i** Tumor volume of xenografts formed after subcutaneous injection of NOD/SCID mice with HT-29 cells infected with an indicated Doxycycline-inducible lentivirus encoding control shRNA or OFD1 shRNA in the presence or absence of Doxycycline in the animal diet. The tumor sizes were measured twice per week. 12 mice per group per time point examined over three independent experiments, two-tailed unpaired student's *t*-test. Data shown represented as mean values ± SD, error bar was defined as SD. **j** Weights of HT-29 xenograft tumors from experiments in (**i**). Data shown represented as mean values ± SD, error bar was defined as SD, $P = 1.3 \times 10^{-9}$, two-tailed unpaired student's *t*-test. **k** Tumor images of indicated HT-29 xenograft tumor genotype from experiments shown in **i**. **l** Tumor volume of xenografts formed after subcutaneous injection of NOD/SCID mice with MDA-MB-231 cells expressing Dox-inducible OFD1 shRNA in the presence or absence of Doxycycline in the animal diet. The tumor sizes were measured twice per week. Eight mice per group per time point. 8 mice per group per time point examined over three independent experiments, two-tailed unpaired student's *t*-test. Data shown represented as mean values ± SD, error bar was defined as SD. **m** Weights of MDA-MB-231 xenograft tumors from experiments in (**l**). Data shown represented as mean values ± SD, error bar was defined as SD, $P = 4 \times 10^{-6}$, two-tailed unpaired student's *t*-test. **n** Tumor images of indicated MDA-MB-231 xenograft tumor genotype from experiments shown in (**l**).

## Plasmid and siRNA transfections

pLive-RFP-LifeAct and pLive-EGFP-OFD1 were transfected into RPE1 cells using Lipofectamine 3000 (Life, L3000015). siRNAs transfections were performed using Lipofectamine RNAiMAX (Life, 13778150). The final concentrations of siRNAs were 30–50 nM. All transfections were performed according to the manufacturer's instructions.

## Lentivirus package and mammalian cell infections

The vectors Lenti-CRISPR-V2, pLV-TO-EGFP-OFD1, pLive-EGFP-Tubulin or pLive-RFP-LifeAct, and pLV-TO-shRNAs were co-transfected with package plasmids psPAX2 (Addgene #12260) and pMD2.g (Addgene #12259) into HEK293 cells. Supernatant medium containing lentivirus was filtered by 0.45 μm filters and then added to target cells with 8 μg/mL polybrene. After 48 h, the infected cells were cultured in a selection medium containing puromycin or blasticidin. The infected cells were selected for 2 weeks and confirmed by immunoblots.

## Protein purification

His-cortactin was purified from DE3 bacteria, as described[41]. The bacterial pellet was resuspended in lysis buffer (20 mM Tris-HCl pH 8.0, 300 mM NaCl, 2% Triton X-100, 10 mM imidazole). After lysis, bacteria debris was removed by centrifugation. The supernatant was purified with nickel-nitrilotriacetic acid (Ni-NTA) metal affinity beads (Qiagen, 30210) and eluted in 20 mM Tris-HCl pH 8.0, 300 mM NaCl, 500 mM imidazole, and 0.05% Tween-20. Eluted fractions were combined and further purified using Mono Q chromatography. OFD1-Flag was expressed in EXPI 293 suspension cells and harvested 48 h post-infection. All purification steps are performed at 4 °C to minimize degradation. The 1 L EXPI 293 suspension cells pellet was thawed and resuspended in 200 mL of Lysis buffer (20 mM Tris-HCl pH 7.5, 150 mM KCl, 1 mM EDTA, 0.1% Brij-97 (Sigma, P6136), protease inhibitor cocktail (APExBIO, K1007)). We applied the solution to the French Pressure Cell and put the cell under the desired pressure (30 to 50 bar). Three passes were required for efficient cell lysis, and cell debris was removed by centrifugation at $3300 \times g$ for 10 min in conical tubes using a swinging bucket rotor. The supernatant was further clarified by ultracentrifugation at $50,000 \times g$ for 1 h in a fixed-angle rotor. The extract was then transferred into four 50 mL tubes containing 50 μL/tube of Flag-M2 agarose resin (Sigma, A2220) pre-equilibrated in Lysis Buffer. The binding reactions were incubated with rotation overnight at 4 °C. Flag-M2 beads were then transferred into a 15 mL tube and then washed three times with 10 mL of Wash buffer-500 (20 mM Tris-HCl pH 7.5, 500 mM KCl, 1 mM EDTA, 0.1% Brij-97) for 10 min, twice with 10 mL of Wash buffer-250 (20 mM Tris-HCl pH 7.5, 250 mM KCl, 1 mM EDTA, 0.1% Brij-97), and three times with 10 mL of Wash buffer-150 (20 mM Tris-HCl pH 7.5, 150 mM KCl, 1 mM EDTA, 0.1% Brij-97). Flag-M2 beads were then transferred into a 1.5 mL centrifuge tube, and several elutions were performed by mixing the 200 μL resin with 200 μL of Flag Elution buffer (20 mM Tris-HCl pH 7.5, 150 mM KCl, 1 mM EGTA, 0.1% Brij-97, 0.2 mg/mL 3 × Flag peptides). Each elution was performed by rocking the slurry (3 h at 4 °C for the first one, 2 h at 4 °C for the second one) and centrifuging at $300 \times g$ for 3 min. The supernatant was further centrifuged at $20,000 \times g$ for 10 min to remove the precipitated protein. Then the elution was flash-frozen in liquid nitrogen and stored at −80 °C.

## Co-immunoprecipitations and immunoblot analysis

For co-immunoprecipitations, transfected HEK293T cells were harvested, washed with chilled PBS, and lysed in TAP lysis buffer for 15 min on ice. Supernatants were collected after $10,000 \times g$ centrifugation at 4 °C. Flag-M2 gel or IgG beads were added into the supernatant and incubated for 2 h at 4 °C. After 3 washes with TAP lysis buffer and 1 wash with elution buffer (without Flag peptides), the bound proteins were eluted with elution buffer containing 200 μg/mL 3×Flag peptides. For immunoblot analysis, the elution products or the cell lysate were denatured with SDS sample buffer, subjected to SDS-PAGE, transferred to 0.22 μm PVDF membranes (Bio-Rad). The PVDF membranes were blocked with 5% milk, incubated with antigen-specific primary antibodies followed by horseradish peroxidase-conjugated secondary antibodies. The signals were visualized with Immobilon Western Chemiluminescent HRP Substrate (Millipore) and detected with X-Ray film. Primary antibodies used in this paper were mouse monoclonal anti-β-tubulin (Developmental Studies Hybridoma Bank E7, 1:10000 dilution), mouse monoclonal anti-GAPDH (Santa Cruz sc-365062, 1:20000 dilution), mouse monoclonal anti-PCM1 (Santa Cruz sc-398365, 1:1000 dilution), mouse monoclonal anti-HA (Sigma-Aldrich H9658, 1:10000 dilution), mouse monoclonal anti-Flag-M2 (Sigma-Aldrich F1804, 1:5000 dilution), mouse monoclonal anti-p21 (Santa Cruz sc-6246, 1:1000 dilution), mouse monoclonal anti-p53 (Santa Cruz sc-126, 1:5000 dilution), mouse monoclonal anti-p27 KIP1 (Cell Signaling Technology 3698, 1:1000 dilution), mouse monoclonal anti-RB1 (Cell Signaling Technology 9309, 1:1000 dilution), rabbit polyclonal anti-IFT20 (Proteintech Group 13615-1-AP, 1:1000 dilution), rabbit

polyclonal anti-RBL1 (Proteintech Group 13354-1-AP, 1:500 dilution), rabbit polyclonal anti-RBL2 (Proteintech Group 27251-1-AP, 1:600 dilution), rabbit polyclonal anti-BBS4 (Proteintech Group 12766-1-AP, 1:1000 dilution)and rabbit polyclonal anti-OFD1 (this paper, 1:5000 dilution)[73]. The HRP secondary antibodies used in this paper were goat anti-mouse IgG, light chain specific (Jackson ImmunoResearch Laboratories 115-035-174, 1:5000 dilution) and mouse anti-rabbit IgG, light chain specific (ImmunoResearch Laboratories 211-032-171, 1:5000 dilution).

## Pyrene-actin polymerization assay

Rabbit skeletal-muscle actin (AKL95), pyrene-labeled muscle actin (AP05), Arp2/3 Protein Complex (RP01P), and GST-VCA protein (VCG03) were purchased from Cytoskeleton Inc. In brief, diluted pyrene-labeled actin and non-labeled actin were added to 10.5 μM with G-actin buffer (5 mM Tris-HCl pH 8.0, 0.2 mM CaCl$_2$, 1 mM TCEP and 0.2 mM ATP) and left on ice for 1 h. The actin was then centrifuged at 100,000 × $g$ at 4 °C in a TLA55 rotor for 1 h. Non-labeled actin was mixed with pyrene-labeled actin and diluted in G-actin buffer (2.625 μM, 20% pyrene labeling) before use. To analyze actin polymerization, the Arp2/3 complex, OFD1, or other testing proteins was added to 100 μL of 1.5× polymerization buffer (10 mM Tris pH 7.5, 75 mM KCl, 3 mM MgCl$_2$, 1.5 mM EGTA, 0.15 mM CaCl$_2$, 0.05% Brij-97, 0.75 mM TCEP, 0.3 mM ATP). Polymerization was initiated by adding 50 μL of pyrene G-actin solution. In some experiments, the reagent and sample volumes were halved. The final concentration of G-actin in the polymerization reaction was 0.875 μM. The pyrene-actin fluorescence signal was monitored at 20-s intervals in a 96-well plate using a VICTOR Nivo fluorometer (Perkin-Elmer) with filters for excitation at 355/40 nm and emission at 405/10 nm. The actin polymerization data was imported into GraphPad Prism and actin polymerization graphs were plotted. The data has been processed as follows: the raw data has been normalized so that the initial background fluorescence values for all experimental conditions are roughly the same, and this data has then been fitted into smooth curves using the Graphpad Prism function 'Nonlin fit:log(agonist) vs. response – Variable slope'.

## RNA isolation and quantitative RT-PCR

Total RNA was isolated from the RPE1 cells using the FastPure Cell/Tissue Total RNA Isolation Kit (Vazyme RC101-01) according to the manufacturer's instructions. 1 μg total RNA was used as the template for a 20 μL reverse transcription reaction using HiScript II Q RT SuperMix (Vazyme R223-01). For quantitative real-time PCR, 20 μg cDNA was used as template for a 10 μL RT-PCR reaction using 2 × SYBR Green Fast qPCR Mix (ABclonal RK21206) on the Applied Biosystems QuantStudio 5 Real-Time PCR instrument. OFD1 relative gene expression was analyzed based on the 2 − ΔΔCt method with GAPDH as an internal control. The primer sequences for real-time qPCR were as follows: for OFD1 primer 1, 5'-ACCAGACGTTTAAGGATCGGG-3' (forward) and 5'-GTTCTCCACTCAATACAGGG TG-3' (reverse); for OFD1 primer 2, 5'-AGCCCAGTCTTTGGCAATAAC (forward) and 5'-GGAG ACGCAGGTTTTCATTTCT-3' (reverse); for GAPDH, 5'-ACAACTTTGG TATCGTGGAA GG-3' (forward) and 5'-GCCATCA CGCCACAGTTTC-3' (reverse).

## Light Microscopy

For immunofluorescence analysis of cells, cells cultured on cover glasses were fixed with 4% paraformaldehyde for 10 min, followed by 10 min of methanol fixation at −20 °C. Cells were incubated for 1 h with primary antibodies, washed twice with PBS, and then incubated with secondary antibodies for 30 min. DNA was visualized by DAPI staining. The stained cells were mounted with Prolong Diamond (Invitrogen). Primary antibodies used in this paper were mouse monoclonal anti-ARL13B (NeuroMab 75-287, 1:1000 dilution), mouse monoclonal anti-Flag-M2 (Sigma-Aldrich F1804, 1:3000 dilution), mouse monoclonal

anti-γ-tubulin (Sigma-Aldrich T6557, 1:2000 dilution), rabbit polyclonal anti-ARL13B (Proteintech Group 17711-1-AP, 1:1000 dilution), rabbit polyclonal anti-OFD1 (1:3000 dilution), rabbit polyclonal anti-Ki67 (Abcam-ab15580, 1:500 dilution), and goat polyclonal anti-CEP164 (Santa Cruz, sc-240226, 1:500 dilution). Secondary antibodies were bovine anti-goat Alexa Fluor 488 (Jackson ImmunoResearch Laboratories, 1:500 dilution), goat anti-mouse Alexa Fluor 488 (Jackson ImmunoResearch Laboratories, 1:500 dilution), goat anti-mouse Alexa Fluor 568 (Life Technologies, 1:300 dilution), goat anti-rabbit Alexa Fluor 594 (Life Technologies, 1:300 dilution) and goat anti-rabbit Alexa Fluor 647 (Life Technologies, 1:300 dilution). Cells were imaged by the Zeiss LSM780 or LSM880 laser scanning confocal microscope. For phalloidin staining, cells were fixed with 4% PFA for 10 min, permeabilized with 0.3% Triton X-100 for 5 min, and stained for 10 min with Phalloidin Alexa Fluor 594 (Life Technologies A12381, 5 units/mL). For live cell imaging, cells were cultured in CO$_2$-independent medium (GIBCO, 18045088) supplemented with 10% FBS (Sigma-Aldrich), 2 mM GlutaMAX (GIBCO, 35050061), and 100 U/mL Penicillin-Streptomycin (GIBCO) at 37 °C. For phase-contrast time-lapse microscopy, images were acquired every 10 min with a Plan-Neofluar 10×/0.3 Ph1 objective (Carl Zeiss), Retiga 2000R camera (Qimaging), on an Axiovert 200 microscope. For fluorescent time-lapse microscopy, images were acquired every 10 s (visualization of F-actin and OFD1) or 5 min (visualization of F-actin and microtubule) with a ×20 or ×40 objective on the Zeiss LSM780 or LSM880 laser scanning confocal microscope. For PFA-PEM fixation, Cells were fixed at 37 °C for 10 min with fresh 4% paraformaldehyde (PFA, TED PELLA, 18505) in the cytoskeleton preserving buffer (PEM) (80 mM PIPES pH 6.8, 5 mM EGTA, 2 mM MgCl$_2$). Cells were permeabilized by PEM with 0.5% Triton-X-100 for 10 min at room temperature, blocked with blocking buffer (5% Bovine Serum Albumin (BSA) in PEM) for 60 min at room temperature, and stained with rabbit polyclonal anti-OFD1 (1:1000 dilution), mouse monoclonal anti-ARP2 (Sigma-Aldrich A6104, 1:200 dilution) in blocking buffer for 60 min at room temperature, followed by goat anti-mouse Alexa Fluor 488 (Jackson ImmunoResearch Laboratories, 1:200 dilution), goat anti-rabbit Alexa Fluor 594 (Life Technologies, 1:200 dilution), Phalloidin Alexa Fluor 488 (Invitrogen A12379, 200 nM), or Phalloidin Alexa Fluor Plus 647 (Invitrogen A30107, 200 nM) in blocking buffer for 30 min at room temperature.

## Fluorescence recovery after photobleaching (FRAP) experiment

FRAP experiments were performed on a Delta Vision OMX SR microscope. Tet-inducible GFP-OFD1-expressing RPE1 cells with 0.1 ng/mL Doxycycline upon DMSO or CK-666 treatment (120 μM, 96 h) were prepared for live cell analysis. Dishes were placed in a 37 °C chamber supplemented with 5% CO$_2$. Images were obtained with a CMOS camera, using the 60 × oil 1.42 objective with transparent oil that has a refractive index of 1.516. Time-lapse images were acquired before and after photobleaching at 1 s per frame for a total time of 10 min (5 s prebleach). For FRAP analysis, GFP-OFD1 condensates were photobleached with a region of interest (ROI) (~8 μm$^2$) at 20% 488 nm laser intensity for 0.5 s. The image-induced photobleaching was corrected by normalizing the fluorescence time course decay in non-bleached regions using the Image J plug-in FRAP Profiler. Normalized FRAP curves were imported into GraphPad Prism.

## Cell cycle analysis by FACS

RPE1 cells were harvested and washed with chilled PBS and then fixed with 75% ethanol at −20 °C overnight. The fixed cells were washed twice with PBS and treated with 100 μg/ml RNase for 30 min. The cells were stained with 40 μg/mL PI and subjected to the BD FACSCalibur™ at the Flow Cytometry Facility of UT Southwestern Medical Center and at the Flow Cytometry Facility of Shanghai Jiao Tong University School of Medicine. Graphical counts for all FACS sequential gating/sorting strategies were described in Supplementary Fig. 7.

## RNA-seq analysis

RNA was purified from RPE1 cells and then reverse-transcribed into cDNA. The RNA-seq libraries were prepared with the NEBNext® Ultra™ Directional RNA Library Prep Kit for Illumina according to the manufacturer's instructions. Index-coded samples were clustered on the cBot Cluster Generation System using the TruSeq PE Cluster Kit v3-cBot-HS (Illumia) according to the manufacturer's instructions. After cluster generation, library preparations were sequenced on the Illumina Novaseq platform and 150 bp paired-end reads were generated by Novogene (Novogene, Tianjin, China). RNA-seq data were aligned to the Ensembl human reference transcriptome (GRCh38, version 94) by HiSat2 and summarized by StringTie as fragments per kilobase transcript per million mapped reads (FPKM). Heatmaps to visualize the data were generated using Excel.

## Mouse tumor xenograft studies

NOD/SCID mice were bred at the animal facility at UT Southwestern Medical Center (UTSW) under specific pathogen-free conditions. All of the mouse experiments were performed according to the guidelines of the Institutional Animal Care and Use Committee (IACUC) at UTSW. The maximal tumor size is limited within 2000 mm³ according to policy of the ethics committee. The tumor sizes were not exceeded 2000 mm³ by volume measurement. Female mice with matched ages were used in each experiment. For the studies of pancreatic cancer and colon cancer, $5 \times 10^6$ PANC-1 cells or $1 \times 10^7$ HT-29 cancer cells were subcutaneously injected into the flank region of seven-week-old female NOD/SCID mice. For the studies of breast cancer, $2 \times 10^7$ MDA-MB-231 cancer cells were orthotopically injected into the mammary gland fat pad of seven-week-old female NOD/SCID mice. The tumor volume was monitored twice a week using the formula (tumor volume = ½ (L × W2)). To study the function of OFD1 in tumor growth, the expression of control or OFD1 shRNA was induced by Doxycycline (500 mg/liter in water and 500 mg/kg in grain-based diet). At the end of the experiments, mice were sacrificed, and tumors were dissected in accordance with institutional guidelines and with approval from the Institutional Animal Care and Use Committee.

## Histology, immunohistochemical (IHC) staining, and TUNEL staining

Xenograft tumors were harvested, and fixed in 4% PFA for 48 h. The fixed samples were embedded in paraffin and sectioned. The IHC staining of paraffin-embedded tissues was performed using OFD1 (this paper, 1:3000 dilution) primary antibodies and the peroxidase Elite ABC-HRP Kit (VECTOR LABORATORIES, PK-6200) according to the manufacturer's instructions. TTF-1 staining in xenografts was classified as strongly positive, moderately positive, or weakly positive using previously described criteria (Saad et al., 2004). For the detection of apoptotic cells, TUNEL staining was performed on tumor xenograft sections according to the manufacturer's instructions (Promega, G3250).

## Clinical human specimens

Colon cancer and lung cancer sections, with corresponding normal tissue microarray (TMA) sections, were prepared by Shanghai Outdo Biotech Co. Ltd. (Shanghai, China). Sample collection and preparation were approved by the Scientific Investigation Board of Taizhou Hospital and were in accordance with the ethical principles originating from the Declaration of Helsinki. These tissue arrays contained tissues from 80 paired colon carcinoma and normal tissue samples together with 20 extra colon carcinoma samples (HColA180Su10-M-069), as well as 60 paired lung cancer and normal tissue samples (HlugC120PT01) were used to examine the expression profiles of OFD1 using immunohistochemistry (IHC). For IHC, TMA sections were incubated with anti-OFD1 antibodies at a 1:4000 dilution for colon cancer samples and a 1:1500 dilution for lung cancer samples. The EnVision+ detection system (Dako) was used per the manufacturer's instructions. IHC stains were scored by two independent pathologists who were blinded to the clinical characteristics of the patients. The scoring system was based on the intensity and extent of staining: staining intensity was classified as 0 (negative), 1 (weak), 2 (moderate), or 3 (strong). Immunostained sections on microarrays were scored by multiplying the intensity (0–3) and area percentage (0–100%) of staining.

## Statistics and reproducibility

For comparing the statistical significance between two groups, the data were analyzed with a two-tailed unpaired student's t-test. Statistical parameters and significance ($P$ value) are reported in the Figures or the Figure Legends. All microscopic, biochemical, and biological assays were independently repeated at least three times.

## Reporting summary

Further information on research design is available in the Nature Portfolio Reporting Summary linked to this article.

## Data availability

All original data that support the findings of this study will be available upon request. The raw data generated in this study including RNA-Seq have been deposited in the database of Gene Expression Omnibus (GEO) under accession code GSE225003. The individual-level data are available under restricted access for download, access can be obtained by authorized access only. The raw individual-level data are protected and are not available due to data privacy laws. The processed individual-level data are available at the database of Gene Expression Omnibus. The individual-level data generated in this study are provided in the Supplementary Information. The individual-level data used in this study are available in the GEO database under accession code GSE225003. Source data are provided with this paper.

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

## Acknowledgements

We thank the Core facility of Basic Medical Science at the Shanghai Jiao Tong University School of Medicine for technical support. We thank Drs. William Snell (University of Maryland), Junmin Pan (Tsinghua University), and Saikat Mukhopadhyay (University of Texas Southwestern Medical Center) for their insightful discussions, Dr. Rolf Brekken (University of Texas Southwestern Medical Center) for technical advice, Andrew Shiau (Ludwig Institute for Cancer Research, San Diego) for kindly providing centrinone, and Meng-Fu Bryan Tsou (Memorial Sloan Kettering Cancer Center) for kindly providing cell lines for preliminary tests. This work was supported, in part, by National Natural Science Foundation of China (91957204, 92254307, 91754205, 31771523) to Q.Z., Mobility program M-0140 to Q.Z., Ministry of Science and Technology of China (2019YFA0508602) to Q.Z., Program of Shanghai Subject Chief Scientist (19XD1402200) to Q.Z., Shanghai Municipal Science and Technology Project (20JC1411100), Ministry of Science and Technology of China (2021YFC2700800) to M.C., National Natural Science Foundation of China (91954123, 31972887) to M.C., Grants from the State Key Laboratory of Oncogenes and Related Genes to M.C., Clinical research projects of Shanghai Municipal Health Commission (20194Y0133) to M.C., National Natural Science Foundation of China (31500627, 32070741) to Z.T., Shanghai Municipal Science and Technology Project (20ZR1430300) to Z.T., NIH (GM096070) to J.S., Welch Foundation (I-1910) to J.S., and Cancer Prevention Research Institute of Texas (CPRIT) grant RP120718 to B.L.. This work was also supported by Shanghai Frontier Science Center of Cellular Homeostasis and Human Diseases and innovative research team of high-level local universities in Shanghai (SHSMU-ZDCX20211800 and SHSMU-ZDCX20211801).

## Author contributions

M.C., X.Z., and C.L. performed most of the biological and biochemical experiments characterizing OFD1 function in mammalian cells and mice. Z.L., N.W., and J.S. performed the cell biology experiments; Z.Z. helped with the mouse experiments; Y.Y. performed the bio-informatics analysis; M.C., B.L., Z.T., J.S., and Q.Z. conceived the concept and supervised the project, M.C., Z.T., and Q.Z. designed the experiments, analyzed the data, and wrote the paper with the help of all authors.

## Competing interests

The authors declare no competing interests.
