## [Peer Review File · Nature Communications]

REVIEWER COMMENTS

Reviewer #1 (Remarks to the Author):

This manuscript describes a role of OFD1 in regulating the actin branching for the restriction point, cytokinesis, and primary ciliogenesis through an interaction with the ARP2/3 complex proteins. OFD1 has a well-established role in left-right asymmetry establishment and skeletal and brain development. However, suppression or promotion of ciliogenesis has been ascribed to the OFD1 depletion (Alfieri et al 2020; Singla et al 2010; Tang et al 2013). Moreover, OFD1 has been reported to be important for cell cycle progression and cell division by regulating the microtubule network (Alfieri et al 2020).

In this manuscript, the authors report that OFD1 depletion promotes ciliogenesis. They also show that OFD1 can interact with the ARP2/3 complex, which also acts as a negative modulator of ciliogenesis. Further analysis reveals that depletion of OFD1 can induce cell quiescence in an RB-dependent way. They also demonstrate that OFD1 regulates the actin filament assembly for the proper cytokinesis and cell cycle progression, which is important for cancer cell growth. Furthermore, the authors show that OFD1-depleted cancer cells develop much smaller tumors in mouse xenograft models.

Overall, this is a very interesting study, which has potentially important implications in understanding how OFD1 regulates the cell cycle and contributes to cancer development. Although OFD1 has been implicated to play roles in the cell cycle through regulation of the microtubule network, this study provides substantial evidence on the role of OFD1 in controlling the actin assembly for the same process. However, more evidence for their mechanistic conclusions is needed to solidify this study.

Major points

1. The major concern relates to the role of OFD1 functioning as a class II NFP to regulate centrosomal actin branching. The authors provide substantial *in vitro* evidence to prove that OFD1 can interact with the ARP2/3 complex and promote actin polymerization with the help of VCA. However, the *in vivo* role of OFD1 requires some further verification. The authors should perform a high-resolution colocalization analysis of OFD1 and ARP2. Moreover, to substantiate the conclusion, the authors should investigate the effects of OFD1 depletion on centrosomal actin branching and the centrosomal accumulation of ARP2 (Farina et al 2016).
2. The authors state that loss of OFD1 largely reduced the F-actin cloud around centrosomes (in line 81) but they present this important result in supplementary figure 1. and no detectable defects in mitotic spindle assembly were observed in OFD1-depleted cells (in line 328). Therefore, proper rescue experiments by using mutants defective in binding ARP2 (OFD1 (Δ 950-1012) and OFD1 7A) are essential to clarify the specificity.
3. Centrosomal actin branching has been reported to be important for centrosomal microtubule nucleation. It's important to clarify whether OFD1 depletion affects the centrosomal microtubule nucleation. In addition, PCM1 is essential for the centrosomal actin assembly (Farina et al 2016). It would be of importance to detect whether the centriolar satellite PCM1 is affected by OFD1 depletion.
4. It would be necessary to show the fission and fusion events by live cell imaging to demonstrate the phase separation of centriolar satellite OFD1. Moreover, centriolar satellite OFD1 condensed into puncta upon the treatment of CK-666 and Cyto D, so it would be important to monitor the EGFP-OFD1 dynamics upon CK-666 or Cyto D treatment by live cell imaging.
5. Although multiple siRNAs targeting OFD1 have been utilized to confirm the specificity, it is essential to test if mutants defective in binding ARP2 (OFD1 (Δ 950-1012) and OFD1 7A) can restore the cell cycle arrest because of their documentation of the OFD1-mediated actin branching surveillance system.

Minor points

1. To make conclusive interpretations of data, quantification of the results from multiple experiments in Fig 1h (weakened protein-protein interaction), Fig 2a (% of ciliated cells), Fig2e-g (decreased/increased protein level), Fig3b (% of cells in G1, S, G2/M phases), Fig 4h (% of ciliated cells), and Fig 5c (% of polyploid cells) would be needed.

2. Flag-OFD1 protein was purified from mammalian EXPI 293 cells under native conditions and used for the in vitro pull-down assay. It's not an appropriate way to distinguish a direct interaction.
3. In line 98, OFD1 is written instead of ofd1.
4. In the method section, the information on how to purify the GST-VCA protein is missing.
5. In the method section, it's unclear what TAP lysis buffer is.
6. The letters of ARP2 should be capitalized.
7. Molecular weight markers are missing in all western blots.
8. Tests for statistical significance are missing in some qualifications.
9. The information on α -Tubulin and CP110 antibodies are missing in the key resource table and the method section.
10. The introduction section should be more comprehensive. To make the manuscript more readable, the authors should re-write the Introduction section to provide sufficient background. In addition, subheadings should be provided in the Results section.
11. In the Method section, Cisplatin and Etoposide are included. Where were these drugs used in this study?

Reviewer #2 (Remarks to the Author):

The manuscript "An Actin Branching network surveillance system controls the restriction point, cytokinesis, and primary ciliogenesis" by Cao et al. is a novel and comprehensive study on OFD1 protein and its role in ciliogenesis and cell division. The authors observed that ciliopathy protein OFD1 coordinates ciliogenesis with actin branching at the centrosome and proliferation. The OFD1-mediated actin branching surveillance system is a comprehensive study with potential interest to basic and clinical scientists. The study adds new components and connections to the established role of actin dynamic in ciliogenesis, and in this regard, the study has some novelty. Although attractive, a study has multiple technical and conceptual flaws, as described below.

The authors completed several in vitro assays to document OFD1's binding to Arp2 and F-actin, supporting its role in actin dynamics. Nevertheless, there is insufficient data to document that OFD1 influences actin branching as no in vitro or in vivo (in cells) branching analyses have been provided. The actin staining around the centrosome is weak and only observed upon overexpression of OFD1. The depletion of OFD1 does not affect the F-Actin structure or assembly, suggesting localization of OFD1 to the F-actin fibers has no biological function. In contrast, localization of OFD1 to branched actin was not documented here (Fig.1). The actin localization of OFD1 in cancer cells was not shown. The co-localization studies with F-actin, including imaging files, lack resolution to define the localization of specificity of association. Overall experimental evidence of low-resolution quality. Parallel experiments with siCTTN, since it seemed to work similar to OFD1 and had been previously shown to affect ciliary dynamics, would be a reasonable control. It is unclear how actin de-branching at the centrosome affects global actin changes in cells upon starvation used as a model of ciliation.

The effects of inhibitors on actin dynamics as in Fig2A show no to limited effects on OFD1 amount or distribution. The ciliation rates need to be quantified and reported. The high-quality 3D projections are required to reconstruct the centrosome region vs. the whole cell to allow for mapping of the OFD1 localization in reference to other actin and centrosome markers. The experimental design for Fig2A has multiple variables complicating the interpretation of the data, including lack of cell cycle synchronization that might affect ciliation and actin dynamics. On top of this, the starvation and inhibitors were applied for 96h, which is significantly above 48-72h of the previously described starvation approach. The nocodazole application for 48h triggers cell death in most normal (non-transformed) cells; the appropriate controls for cell viability are needed. The morphological observations of limited liquid phase diffusion are not well supported. The interpretation of the data in Fig,2bc is based on OFD1 over-expression studies. Moreover, as recently was published, the 1,6-hexanediol renders both kinases and phosphatases virtually inactive, suggesting the observed effects might have little to do with actin. Immunofluorescence images in panel A do not support the conclusion that CK-666 treatment leads to OFD1 reduction. The proliferation assays with K67 must be performed on synchronized cells to allow for similar conditions. The HD-high density cultures are difficult to interpret since siOFD1 cells do not

proliferate (Fig.3). How were these cultures produced in the first place? The RNAseq data in Fig3c needs to show control cells, biological replicas, and statistics. The OFT20 and CEP164-/cell-based studies have not been performed on synchronized cells. The defects of CEP164 are well described in cancer cells, and depletion causes hyper-proliferation. In this regard, the current data contradict some published data. The p53 KO and Tag data in RPE cells lack the controls for the basal levels of proliferation and cell death. The conclusion that Tag in RPE cells somehow allows bypassing G0 in OFD1 KO lacks experimental evidence. To clarify the effects of OFD1 on cilia vs. actin dynamics around centrosome, rescue experiments with OFD1 point mutants at the Arp2 binding site in CK-666 treated cells are needed. Current data does not uncouple the generic effect of CK-666 on actin dynamics in cells from the centrosome/actin interface and thus might be indirect effects. The data in Fig4f-g contradict the original premise as cells with depletion of RB1\RB11\RBL2 have 80% ciliated cells entering the cell cycle and Ki67positive (S phase). The RB pathway also directly influences actin dynamics via ILK and mDia1. The drastic decrease in proliferation upon co-expression of hTERT and SV40 Tag was previously described and associated with significant cell death with higher basal levels of divisions and ki67 positivity. The authors called these cells transformed without evidence to support malignant transformation. The role of OFD1 in cancer is controversial, and the data shown in Extended Fig.6c-e requires more depth on how the panels were generated with specific database links and statistics. The analysis of OFD1 expression in protein atlas <https://www.proteinatlas.org/ENSG00000046651-OFD1/pathology> or <https://kmplot.com/analysis/> shows that for some cancers up-regulation of OFD1 is a favorable prognostic marker and few cancers have a significant difference in OS (overall survival) based OFD1 expression.

The overall study is comprehensive and provided evidence to support the notion that centrosome-associated OFD1 is essential for actin accumulation and control of ciliary dynamics. However, its role in cell proliferation and mitosis seems more complicated and potentially actin independent.

Reviewer #3 (Remarks to the Author):

In this manuscript, Cao and his colleagues describe a novel role of OFD1 in cell cycle control through regulating actin branching at the centrosome region, which affects several cellular events including ciliogenesis, cytokinesis and cell cycle restriction point. They first identified OFD1 as a novel Class II actin nucleation promoting factor which promotes actin branching around the centrosome. They further confirmed that its NPF function is important for interphase ciliogenesis and mitotic spindle assembly. Then they proved that OFD1 inactivation through siRNA or shRNA led to cell cycle arrest at G0 and ciliogenesis through the RB-dependent pathway. Intriguingly, they found that while OFD1 knockdown led to cell cycle arrest in normal cells, its inhibition greatly suppressed cancer cell proliferation and tumor growth. These results shed light on centrosome-actin, actin-ciliogenesis, actin-mitosis relationships and cancer therapies. Overall, this is a well-designed project with sufficient evidence to support the conclusions and should be published.

Minors and suggestions

1. Fig. 4f-4i showed that co-KD of RB1/L1/L2 could rescue cell cycle arrest caused by siOFD1, but did not cause cilia disassembly, suggesting OFD1 may have a direct role in ciliogenesis and cilia disassembly. Are the cilia length the same when co-KD RB proteins compare to OFD1 KD alone?
2. Actin filament destabilization by CK666 or cytoD caused centriolar satellite proteins relocation. Does the actin network function in centriolar satellite maintenance? This need to be further discussed.
3. Line 324, CEP164-/- cells also showed inhibited proliferation should add "when OFD1 is depleted" to be more accurate.
4. OFD1 KD affects actin branching around the centrosome and Arp2/3 inhibition by CK666 treatment inhibits actin branching more broadly. Did the authors directly compare the ability of OFD1 KD and CK666 in inducing ciliation in the presence of serum?

5. Images in Fig.2a are small. Showing zoomed crop regions like Fig.2c could help to illustrate the localization changes of OFD1.

6. In Tet-on/off systems, Dox concentrations are typically used at 100 nm to 1 um. Here the authors described a Dox concentration as low as 0.01 nm. I wonder if this is a typo.

Point-by-point response to reviewer's comments:

Reviewer #1 (Remarks to the Author):

Major points

1. The major concern relates to the role of OFD1 functioning as a class II NFP to regulate centrosomal actin branching. The authors provide substantial *in vitro* evidence to prove that OFD1 can interact with the ARP2/3 complex and promote actin polymerization with the help of VCA. However, the *in vivo* role of OFD1 requires some further verification. The authors should perform a high-resolution co-localization analysis of OFD1 and ARP2. Moreover, to substantiate the conclusion, the authors should investigate the effects of OFD1 depletion on centrosomal actin branching and the centrosomal accumulation of ARP2 (Farina et al 2016).

Response: Great suggestions. To address the reviewer's concern, we have attempted different methods to investigate the function of OFD1 *in vivo*. Using an improved fixation method for actin staining, 37°C for 10 min with fresh 4% paraformaldehyde (PFA) in the cytoskeleton preserving buffer (PEM) (80 mM PIPES pH 6.8, 5 mM EGTA, 2 mM MgCl₂) (Leyton-Puig, Biol Open, 2016, PMID: 27378434; Pereira, Front Immunol, 2019, PMID: 31024536), we demonstrated that OFD1 plays an important role in the centrosomal localization of Arp2/3 complex and the complex-mediated centrosomal actin branching. As shown in **Fig. 1e** (Response Fig. 1a), endogenous ARP2 co-localized with OFD1, while the centrosomal population of ARP2 dramatically reduced upon OFD1 depletion. Meanwhile, in RPE1 (**Supplementary Fig. 1g**, Response Figure 1b) and HeLa cells (**Fig. 1f**, Response Fig. 1c), the centrosome was surrounded by a highly dynamic F-actin network, which was compromised upon OFD1 depletion. HeLa cells have a relatively low background of cortical actin, which makes them a better model system for observing centrosomal branching actin network (Farina, 2019 EMBO, PMID:31015335). We improved the resolution by using Zeiss LSM 880 Microscope equipped with an Airyscan module which presented greatly improved images in the revised manuscript.

Response Fig. 1. OFD1 facilitates centrosomal actin branching. **a** Representative imaging of endogenous ARP2 or OFD1 in RPE1 cells transiently transfected with control or OFD1 siRNA for 72 hours and after fixation with PFA-PEM (Left Panel). ARP2 fluorescence integrated over a 1- μ m-diameter circle around the centrosome for si-Control or si-OFD1 condition, **** $P < 0.0001$, two-tailed unpaired student's t -test (Right Panel). **b** Representative imaging of endogenous OFD1 (green) and F-actin (phalloidin, red, Gamma-adjusted (0.5)) in RPE1 cells transiently transfected with control or OFD1 siRNA for 72 hours and after fixation with PFA-PEM (Left Panel). F-actin fluorescence integrated over a 3- μ m-diameter circle around the centrosome for si-Control or si-OFD1 condition, **** $P < 0.0001$, two-tailed unpaired student's t -test (Right Panel). **c** Representative imaging of endogenous OFD1 (green) and F-actin (phalloidin, red, Gamma-adjusted (0.5)) in HeLa cells transiently transfected with control or OFD1 siRNA for 72 hours and after fixation with PFA-PEM (Left Panel). F-actin fluorescence integrated over a 3- μ m-diameter circle around the centrosome for si-Control or si-OFD1 condition, **** $P < 0.0001$, two-tailed unpaired student's t -test (Right Panel).

2. The authors state that loss of OFD1 largely reduced the F-actin cloud around centrosomes (in line 81) but they present this important result in Extended data figure 1. and no detectable defects in mitotic spindle assembly were observed in OFD1-depleted cells (in line 328). Therefore, proper rescue experiments by using mutants defective in binding ARP2 (OFD1 (Δ 950-1012) and OFD1 7A) are essential to clarify the specificity.

Response: Good suggestion. In the revised version, as we responded in the previous question, we carried out super-resolution imaging and assessed the function of OFD1 to regulate the centrosomal actin branching *in vivo*, and this part of data is now included in Fig. 1e (Response Fig. 1a), Supplementary Fig. 1g (Response Fig. 1b) and Fig. 1f (Response Fig. 1c). Furthermore, we also evaluated the rescue functions of siRNA-resistant EGFP-OFD1 and EGFP-OFD1-W1012A mutant, which is more specific than OFD1 (Δ 950-1012) and OFD1 7A, by comparing the centrosomal F-actin. The expression of the EGFP-OFD1 rescued OFD1 depletion-induced centrosomal F-actin clouds reduction in cells. In contrast, the expression of the EGFP-OFD1-W1012A mutant failed to do so (Supplementary Fig. 1j, Response Fig. 2).

Response Fig. 2. The effects of OFD1 and its W1012A mutant on centrosomal actin branching. a Representative imaging of EGFP-OFD1 or EGFP-OFD1(W1012A) and F-actin (phalloidin, red, Gamma-adjusted (0.5)) in Tet-inducible EGFP-OFD1-expressing or EGFP-OFD1(W1012A)-expressing HeLa cells treated with 0.1 ng/mL Doxycycline and OFD1 siRNA for 72 hours and after fixation with PFA-PEM (Left Panel). F-actin fluorescence integrated over a 3- μ m-diameter circle around the centrosome for EGFP-OFD1 or EGFP-OFD1(W1012A) condition, **** $P < 0.0001$, two-tailed unpaired student's t-test (Right Panel).

3. Centrosomal actin branching has been reported to be important for centrosomal microtubule nucleation. It's important to clarify whether OFD1 depletion affects the centrosomal microtubule nucleation. In addition, PCM1 is essential for the centrosomal actin assembly (Farina et al 2016). It would be of importance to detect whether the centriolar satellite PCM1 is affected by OFD1 depletion.

Response: Per the reviewer's suggestion, we compared the centrosomal microtubule assembly in control and OFD1-depleted cells. As shown in the Response Fig. 3a, no obvious difference of microtubule nucleation was observed. The depletion of OFD1 did not significantly affect the centrosomal localization of the γ -tubulin complex, which is essential for microtubule nucleation. Multiple studies have demonstrated that the localization of proteins to the centriolar satellites was mutually dependent. Consistent with previous studies (Lopes, J Cell Sci. 2011, PMID: 21266464), the recruitment of PCM1 to the centriolar satellite was reduced upon OFD1 depletion (Fig. 2a and Supplementary Fig.2b, Response Fig. 3b), though the protein level of PCM1 in whole cells remained unchanged (Supplementary Fig. 2c, Response Fig. 3c).

Response Fig. 3. Centrosomal microtubule nucleation or PCM1 localization upon OFD1 depletion. a Representative image of microtubules (Green) and centrosomes (Red) in RPE1 cells transiently transfected with control or OFD1 siRNA and quantification of the intensity of microtubule around centrosomes. Upon treated with nocodazole for 4 hours to disassemble microtubules, nocodazole was washed out to allow microtubule nucleation for 4 minutes. **b** hTERT-RPE1 cells were co-stained with antibodies against OFD1 and PCM1. Cells were subjected to DMSO, serum starvation, 120 μ M CK-689, 120 μ M CK-666, 100 nM

Cyto D, 20 μ M SMIFH2 treatment for 96 hours, si-Control (control siRNA), si-OFD1 (siRNA targeted OFD1) treatment for 72 hours or 15 ng/mL nocodazole treatment for 48 hours. Scale bars: 10 μ m, 3 μ m (zoom in, right). **c** Immunoblot analysis of the protein levels of OFD1 and two other proteins known as satellite components, PCM1 and BBS4, were performed on samples from hTERT-RPE1 cells of indicated conditions. Cells were pretreated with DMSO, 120 μ M CK-666, or 120 μ M CK-689 for 96 hours before being treated with 20 μ M CQ for 9 hours or 1 μ M MG132 for 6 hours.

4. It would be necessary to show the fission and fusion events by live cell imaging to demonstrate the phase separation of centriolar satellite OFD1. Moreover, centriolar satellite OFD1 condensed into puncta upon the treatment of CK-666 and Cyto D, so it would be important to monitor the EGFP-OFD1 dynamics upon CK-666 or Cyto D treatment by live cell imaging.

Response: Thanks for the comments. We carried out live cell imaging to observe the phase separation events of centrosomal OFD1. In control cells, the centrosomal EGFP-OFD1 was much more dynamic with high frequency of fission and fusion events, which indicated that OFD1 underwent liquid-to-liquid phase separation. However, in the CK-666-treated cells, the fission and fusion of EGFP-OFD1 were largely compromised, suggesting a liquid-to-gel transition (Fig. 2b, Response Fig. 4).

a

Response Fig. 4. The phase separation events of OFD1. a Live cell images of the fusion and fission events of centrosomal OFD1 in Tet-inducible EGFP-OFD1-expressing RPE1 cells treated with 0.1 ng/mL Doxycycline and DMSO or 120 μ M CK-666 for 72 hours.

5. Although multiple siRNAs targeting OFD1 have been utilized to confirm the specificity, it is essential to test if mutants defective in binding ARP2 (OFD1 (Δ 950-1012) and OFD1 7A) can restore the cell cycle arrest because of their documentation of the OFD1-mediated actin branching surveillance system.

Response: Thanks for the comments. To test the functions of different OFD1 variants in cell cycle regulation, we generated siRNA-resistant Doxycycline-inducible OFD1-WT and OFD1-W1012A mutant RPE1 cells. Upon depletion of endogenous OFD1, the expression of OFD1-WT at appropriate levels rescued OFD1 depletion-induced quiescence and ciliogenesis, while the expression of OFD1-W1012A mutant failed to do so (Fig. 4a, b and Supplementary Fig. 4f-h, Response Fig. 5).

Response Fig. 5. OFD1 regulates cell cycle and ciliogenesis via controlling centrosomal actin branching. **a** Representative images of ARL13B and Ki67 staining in Tet-inducible EGFP-OFD1-expressing RPE1 cells with indicated titration of Doxycycline and indicated siRNAs for 72 hours. **b** Quantitation of cells positive for ARL13B or Ki67 staining in **a**. Data shown represent mean \pm SD percentage of cells from triplicate samples. $***P < 0.001$, two-tailed unpaired student's *t*-test. **c** Representative images of ARL13B and Ki67 staining in Tet-inducible EGFP-OFD1-W1012A-expressing RPE1 cells with indicated titration of Doxycycline and indicated siRNAs for 72 hours. Cilia are marked by arrowheads. **d** Quantitation of cells positive for ARL13B or Ki67 staining in **c**. Data shown represent mean \pm SD percentage of cells from triplicate samples. $***P < 0.001$, two-tailed unpaired student's *t*-test. **e**

Immunoblot analysis for GFP and α -tubulin of Tet-inducible EGFP-OFD1 or EGFP-OFD1-W1012A expressing RPE1 cells with indicated titration of Doxycycline and indicated siRNAs. The numbers under the gel lanes represent the ratio of EGFP-OFD1 band intensity to α -tubulin band intensity, which were normalized relative to the line 5 sample.

Minor points

1. To make conclusive interpretations of data, quantification of the results from multiple experiments in Fig 1h (weakened protein-protein interaction), Fig 2a (% of ciliated cells), Fig 2e-g (decreased/increased protein level), Fig 3b (% of cells in G1, S, G2/M phases), Fig 4h (% of ciliated cells), and Fig 5c (% of polyploid cells) would be needed.

Response: Thanks for the suggestion. All mentioned quantitative data were included in the revised figures.

2. Flag-OFD1 protein was purified from mammalian EXPI 293 cells under native conditions and used for the in vitro pull-down assay. It's not an appropriate way to distinguish a direct interaction.

Response: Thanks for the comments. The Flag-OFD1 expressed in mammalian cells was purified with high-salt washing to disrupt protein-protein interaction and eliminate possible contamination. My lab is experienced in purification of proteins from mammalian cells, Flag-M2 bead interaction could be resistant to high-salt (0.5 M) and detergent wash but not interacting proteins or contaminants. As shown in **Supplementary Fig. 1a**, the purity of Flag-OFD1 was suitable for the test of interaction. Please refer the more detailed information from Methods.

3. In line 98, OFD1 is written instead of ofd1.

Response: Thanks for the suggestion. We corrected the mistake.

4. In the method section, the information on how to purify the GST-VCA protein is missing.

Response: The GST-VCA is commercially available. We included the commercial information in the method section.

5. In the method section, it's unclear what TAP lysis buffer is.

Response: Thanks for the comments. The TAP (Tandem affinity purification) buffer is a buffer used for protein complex purification, which contains 20 mM Tris HCl (pH 7.5), 150 mM NaCl, 0.5% Nonidet P-40, 1 mM NaF, 1 mM Na₃VO₄, 1 mM EDTA, protease inhibitor mixture (Roche). At this salt concentration (150 mM), most of protein-protein interactions are reserved. If the salt concentration of the wash buffer raised to 500 mM (for protein purification), essentially all protein-protein interactions will be disrupted. Using the TAP buffer, we successfully identified multiple protein complexes and demonstrated their function in several publications (Sun, PNAS, 2008, PMID:19050071; Chen, Mol Cell. 2012, PMID:22342342; Tang, Nature, 2013, PMID:24089205). We added detailed information in the Methods

6. The letters of ARP2 should be capitalized.

Response: Thanks for the suggestion. We revised this as the reviewer suggested.

7. Molecular weight markers are missing in all western blots.

Response: Thanks for the comments. In the revised version, we add the Molecular weight markers for all blots.

8. Tests for statistical significance are missing in some qualifications.

Response: Thanks for the suggestion. We added the statistical significance to the quantifications.

9. The information on α -Tubulin and CP110 antibodies are missing in the key resource table and the method section.

Response: Thanks for the comments. We added the information of α -Tubulin and CP110 antibodies to the key resource table and the method section.

10. The introduction section should be more comprehensive. To make the manuscript more readable, the authors should re-write the Introduction section to provide sufficient background. In addition, subheadings should be provided in the Results section.

Response: Thanks for the suggestion. We revised these parts in the manuscript.

11. In the Method section, Cisplatin and Etoposide are included. Where were these drugs used in this study?

Response: Good eyes. We did not use these drugs and have deleted the information.

Reviewer #2:

The manuscript “An Actin Branching network surveillance system controls the restriction point, cytokinesis, and primary ciliogenesis” by Cao et al. is a novel and comprehensive study on OFD1 protein and its role in ciliogenesis and cell division. The authors observed that ciliopathy protein OFD1 coordinates ciliogenesis with actin branching at the centrosome and proliferation. The OFD1-mediated actin branching surveillance system is a comprehensive study with potential interest to basic and clinical scientists. The study adds new components and connections to the established role of actin dynamic in ciliogenesis, and in this regard, the study has some novelty. Although attractive, a study has multiple technical and conceptual flaws, as described below.

The authors completed several *in vitro* assays to document OFD1’s binding to Arp2 and F-actin, supporting its role in actin dynamics. Nevertheless, there is insufficient data to document that OFD1 influences actin branching as no *in vitro* or *in vivo* (in cells) branching analyses have been provided.

Response: Thanks for the comments. The Arp2/3 complex promotes actin polymerization by generating new actin filaments that branch-off from the side of pre-existing filaments at a 70° angle to form a Y-branched network. Several published studies have shown that centrosome-associated F-actin is branching actin that nucleated with the centrosomal localized Arp2/3 complex (Kim, Nature, 2010, PMID: 20393563; Farina, Nat Cell Biol, 2016, PMID: 26655833; Obino, Nat Commun. 2016, PMID: 26987298; Inoue, EMBO, 2019, PMID: 30902847). In this study, we found that centrosomal protein, OFD1 interacts with Arp2/3 complex and F-actin, functions as a newly identified class II NFP, do not promote actin branching vigorously alone, but synergistically facilitates Arp2/3 mediated actin polymerization and actin branching in the presence of VCA (the functional domain of class I NFPs) using the pyrene actin polymerization assay in **Fig.1i and Supplementary Fig 1d, e**. These data strongly support that OFD1 promotes Arp2/3 mediated actin branching *in vitro*.

We agree with the reviewer that it is important to provide more evidence that OFD1 influences actin branching *in vivo*. In the revised manuscript, as we responded to the reviewer 1, we carried out super-resolution imaging and assessed the function of OFD1 in centrosomal actin branching *in vivo*, in RPE1 (**Supplementary Fig. 1g**, Response Fig. 1b) and HeLa cells (**Fig. 1f**, Response Fig. 1c), the centrosomes were surrounded by a highly dynamic F-actin network, which was compromised upon OFD1 depletion. HeLa cells have a relatively low background of cortical actin, which makes them a better model system for observing centrosomal branching actin network (Farina, EMBO, 2019, PMID:31015335). We improved the resolution by using Zeiss LSM 880 Microscope equipped with an Airyscan module which presented greatly improved images in the revised manuscript. We believe these newly added data strongly support that OFD1 influences actin branching surrounding centrosome *in vivo*.

Response Fig. 1. OFD1 facilitates centrosomal actin branching. **a** Representative imaging of endogenous ARP2 or OFD1 in RPE1 cells transiently transfected with control or OFD1 siRNA for 72 hours and after fixation with PFA-PEM (Left Panel). ARP2 fluorescence integrated over a 1- μ m-diameter circle around the centrosome for si-Control or si-OFD1 condition, **** $P < 0.0001$, two-tailed unpaired student's t -test (Right Panel). **b** Representative imaging of endogenous OFD1 (green) and F-actin (phalloidin, red, Gamma-adjusted (0.5)) in RPE1 cells transiently transfected with control or OFD1 siRNA for 72 hours and after fixation with PFA-PEM (Left Panel). F-actin fluorescence integrated over a 3- μ m-diameter circle around the centrosome for si-Control or si-OFD1 condition, **** $P < 0.0001$, two-tailed unpaired student's t -test (Right Panel). **c** Representative imaging of endogenous OFD1 (green) and F-actin (phalloidin, red, Gamma-adjusted (0.5)) in HeLa cells transiently transfected with control or OFD1 siRNA for 72 hours and after fixation with PFA-PEM (Left Panel). F-actin fluorescence integrated over a 3- μ m-diameter circle around the centrosome for si-Control or si-OFD1 condition, **** $P < 0.0001$, two-tailed unpaired student's t -test (Right Panel).

The actin staining around the centrosome is weak and only observed upon overexpression of OFD1. The depletion of OFD1 does not affect the F-Actin structure or assembly, suggesting localization of OFD1 to the F-actin fibers has no biological function. In contrast, localization of OFD1 to branched actin was not documented here (Fig.1). The actin localization of OFD1 in cancer cells was not shown. The co-localization studies with F-actin, including imaging files, lack resolution to define the localization of specificity of association. Overall experimental evidence of low-resolution quality.

Response: Thanks for the comments. The F-actin structure around the centrosome was high-dynamic branching actin. The observation of OFD1 at branched actin is very challenging. We have attempted different methods to capture the colocalization of OFD1 with the Arp2/3 protein complex. Using an improved fixation method for actin staining, 37°C for 10 min with fresh 4% paraformaldehyde (PFA) in the cytoskeleton preserving buffer (PEM) (80 mM PIPES pH 6.8, 5 mM EGTA, 2 mM MgCl₂) (Leyton-Puig, Biol Open, 2016, PMID: 27378434; Pereira, Front Immunol, 2019, PMID: 31024536), we could observe the colocalization of OFD1 and ARP2 at the centrosomal region. As shown in Fig. 1e (Response Fig. 1a), endogenous ARP2 co-localized with OFD1, while the centrosomal population of ARP2 dramatically reduced upon OFD1 depletion. Meanwhile, in RPE1 (Supplementary Fig. 1g, Response Fig. 1b) and a cancer cell line - HeLa cells (Fig. 1f, Response Fig. 1c), the centrosomes were surrounded by a highly dynamic F-actin network, which was compromised upon OFD1 depletion. HeLa cells have a relatively low background of cortical actin, which makes them a better model system for observing centrosomal branching actin network (Farina, EMBO, 2019, PMID:31015335). We improved the resolution by using Zeiss LSM 880 Microscope equipped with an Airyscan module which presented greatly improved images in the revised manuscript.

Parallel experiments with siCTTN, since it seemed to work similar to OFD1 and had been previously shown to affect ciliary dynamics, would be a reasonable control.

Response: Great suggestion. CTTN is a Class II NPF that functions globally in the cell. Per the reviewer's suggestion, we included the siCTTN as a control. Similar to the cells depleted OFD1, cells treated with CTTN siRNA also entered into quiescence with cilia formation (Response Fig. 6).

Response Fig. 6. CTTN and OFD1 depletion on cell cycle and ciliogenesis. a Representative images of ARL13B and Ki67 staining of RPE1 cells transfected with control siRNA, OFD1 siRNA or CTTN siRNA for 72 hours. Cilia are marked by arrowheads (Left Panel). Quantitative data for the staining (Right Panel).

It is unclear how actin de-branching at the centrosome affects global actin changes in cells upon starvation used as a model of ciliation.

Response: Thanks for the comments. A growing list of evidence has shown that actin de-branching at the centrosomes promotes ciliogenesis. Tracing back to 2010, Kim and colleagues carried out a functional genomic screen using RNA interference (RNAi) to identify human genes involved in ciliogenesis, and identified regulators of actin dynamic affect cilia formation (Kim, Nature, 2010, PMID: 20393563). Further investigation demonstrated that blocking actin assembly facilitates ciliogenesis by stabilizing the pericentrosomal preciliary compartment (PPC), a previously uncharacterized compact vesiculotubular structure storing transmembrane proteins destined for cilia during the early phase of ciliogenesis (Kim, Nature, 2010, PMID: 20393563). In 2012, researchers from Zhu Lab, identified the microRNA, miR-129-3p, which controlled cilia formation by repressing branched F-actin (Cao, Nat Cell Biol, 2012, PMID: 22684256). Several studies also suggested that inhibition of the centrosomal actin network may facilitate the transportation ciliary vesicle to the basal body or the release of actin-binding protein into cilia (Malicki, Trends Cell Biol. 2017, PMID: 27634431; Kim, Nature, 2010, PMID: 20393563; Obino, Nat Commun. 2016, PMID: 26987298) However, how the centrosomal actin is appropriately regulated during ciliogenesis and its underlying mechanism remain elusive. In this study, we propose that OFD1 plays a crucial role in ciliogenesis by regulating the centrosomal actin network. Although serum starvation indeed causes global actin dynamic changes (Chrzanowska-Wodnicka, J Cell Sci, 1994, PMID: 7706413), we did not emphasize that actin de-branching at centrosome affected global actin dynamics upon serum starvation, on the other hand, we believe that OFD1 affects local actin network surrounding centrosome to regulate ciliogenesis.

The effects of inhibitors on actin dynamics as in Fig2A show no to limited effects on OFD1 amount or distribution. The ciliation rates need to be quantified and reported. The high-quality 3D projections are required to reconstruct the centrosome region vs. the whole cell to allow for mapping of the OFD1 localization in reference to other actin and centrosome markers. The experimental design for Fig2A has multiple variables complicating the interpretation of the data, including lack of cell cycle synchronization that might affect ciliation and actin dynamics. On top of this, the starvation and inhibitors were applied for 96h, which is significantly above 48-72h of the previously described starvation approach.

Response: Thanks for the suggestion. To address the reviewer's concern, we synchronized cells by serum starvation for 72 hours before the treatment of actin inhibitors. We performed FACS to analyze the efficiency of synchronization (Supplementary Fig. 2c, Response Fig. 7a). As shown in the response figure, cells remained efficiently synchronized after release and moved homogeneously through the cell cycle for 24 hours (Supplementary Fig. 2c, Response Fig. 7a). The percentage of ciliated cells and the percentage of Ki67-positive cells were quantified after the release accompanied by application of actin inhibitors treated for 72 hours (Supplementary Fig. 2e, Response Fig. 7b). We also observed the localization of endogenous OFD1 upon treatment of actin inhibitors and the well-established centriolar satellites marker PCMI1 was included as a control. The amount of OFD1 at centrosomes was quantified using the SUM z projection

method on z-stacks of images in Image J (Supplementary Fig. 2d, Response Fig. 7c, d). The high-quality 3D projection images of these z-stacks indicate that OFD1 redistributes from discrete puncta into a few large centriolar aggregates under the treatment of actin inhibitors (Supplementary Fig. 2d, Response Fig. 7c, d). We also reconstructed the high-quality 3D projections of the whole cell and the centrosome region of HeLa cells to allow for mapping of the OFD1 localization in the F-actin network (Response Fig. 7e).

Response Fig. 7. Actin debranching phenocopies OFD1 depletion and OFD1 is dynamically regulated during actin debranching. **a** hTERT-RPE1 cells were serum-starved for 72h to synchronize cells and released (time 0) for increasing time periods in fresh medium with serum as indicated. Cells were analyzed by flow cytometry to calculate the percentage of cells in each phases of the cell cycle. **b** Representative images of Ki67 and ARL13B staining in Synchronized hTERT-RPE1 treated with indicated siRNA, 120 μ M CK-689, 120 μ M CK-666 or 100 nM Cyto D for 72h. Cilia are marked by arrowheads (Left Panel). Quantitative data for the staining (Right Panel). Data shown represent mean \pm SD percentage of cells from triplicate samples. **** $P < 0.0001$, two-tailed unpaired student's *t*-test. **c** Representative the SUM Z projection images of OFD1 and PCM1 staining in Synchronized hTERT-RPE1 treated with indicated siRNA, 120 μ M CK-689, 120 μ M CK-666 or 100 nM Cyto D for 72h (Left Panel). Quantitative data of OFD1 intensity for the staining (Right Panel), **** $P < 0.0001$, two-tailed unpaired student's *t*-test. **d** 3D projection images of OFD1 and PCM1 staining in Synchronized hTERT-RPE1 treated with indicated siRNA, 120 μ M CK-689, 120 μ M CK-666 or 100 nM Cyto D for 72h. **e** 3D projection images of OFD1(green) and F-actin (phalloidin, red, Gamma-adjusted (0.5)) or PCM1 (red) staining in HeLa cells.

The nocodazole application for 48h triggers cell death in most normal (non-transformed) cells; the appropriate controls for cell viability are needed.

Response: Thanks for the comments. High concentrations (50-100 ng/mL) of nocodazole might induce cell death according to published studies (Signoretto, Cell Physiol Biochem. 2016, PMID: 26824457; Li, J Radiat Res. 2011, PMID: 21785236). In this study, we used low concentration of nocodazole at 15 ng/mL in our experiments. To verify the cell viability under treatment, we carried out FACS analysis to indicate apoptotic cell death. As shown in the Response Fig. 8, we observed no obvious cell death in RPE1 cells treated with 15 or 30 ng/mL for 48 hours. Compared to the nocodazole treatment, the apoptosis inducer amonafide, which was included as a control, induced strong cell death (Response Fig. 8).

Response Fig. 8. Cell viability upon low concentrations of nocodazole. **a** Detection of cell death in RPE1 cells by Annexin V-PI staining assay. Cells were treated with indicated DMSO, 20 μ M amonafide, 15 ng/ml nocodazole, 30 ng/ml nocodazole for 48h (Upper Panel). Cell morphology was examined by a bright-field microscope (Lower Panel).

The morphological observations of limited liquid phase diffusion are not well supported. The interpretation of the data in Fig.2bc is based on OFD1 over-expression studies. Moreover, as recently was published, the 1,6-hexanediol renders both kinases and phosphatases virtually inactive, suggesting the observed effects might have little to do with actin. Immunofluorescence images in panel A do not support the conclusion that CK-666 treatment leads to OFD1 reduction.

Response: Thanks for the comments. For the phase separation experiments, we not only examined EGFP tagged OFD1 but also endogenous OFD1 and other centrosomal proteins in response to actin dynamic changes, as shown in Fig. 2a (Response Fig. 9a) and the Supplementary Fig. 2a, b. In the indicated conditions, at least a population of OFD1 condensed at centriolar satellites, and the other pool was degraded.

Recent publication demonstrated that centrosomal proteins undergo liquid-to-liquid phase separation to generate amorphous aggregates that are capable of undergoing dynamic turnover and inter-aggregate fusion. And an increasing list of evidence suggests that the ability of these proteins to phase separate from the cytosol is important for their function. We studied how actin depolymerization affected OFD1 aggregation by comprehensive approaches, including the photo bleach assay, fission and fusion assay, and 1,6-hexanediol disassembly assay. The photo bleach assay proved that the dynamics of OFD1 were reduced when actin branching was inhibited (Fig. 2c). We also observed, OFD1 aggregates undergo fusion and fission when treated with or without CK-666 (Fig. 2b, videos 3-5, Response Fig. 9b). OFD1 aggregates disappeared when treated with a widely used phase separation condensates disruptor 1,6-hexanediol. We understand the concern of the reviewer that 1,6-hexanediol treatment may not be valid to support this conclusion since it might affect kinases and phosphatases activity which might indirectly affect phase separation. We will remove this part of data using 1,6-hexanediol. Nevertheless, our conclusion based on the photo-bleaching and fission-and-fusion experiments strongly support that OFD1 underwent liquid-liquid phase separation under normal condition, however when actin was debranched, a population of OFD1 experienced liquid-to-gel-like aggregation.

For the immunofluorescence images in panel A, we only chose the most representative images to illustrate the size and distribution of OFD1 punctuate (Fig. 2a, Response Fig. 9a). To better quantify the intensity of the OFD1 puncta, we quantified the intensity and amount of OFD1 at centrosomes in a group of cells by using the SUM z projection method on z-stacks of images in Image J. After normalization and quantification, we clearly observed that CK-666 treatment led to OFD1 reduction (Supplementary Fig. 2d, Response Fig. 9c). This is also consistent with the results of Western blot (Fig. 2e and Supplementary Fig. 2f).

Response Fig. 9. OFD1 is dynamically regulated during actin debranching. **a** hTERT-RPE1 cells were co-stained with antibodies against OFD1 and PCM1. Cells were subjected to DMSO, serum starvation, 120 μM CK-689, 120 μM CK-666, 100 nM Cyto D, 20 μM SMIFH2 treatment for 96 hours, si-Control (control siRNA), si-OFD1 (siRNA targeted OFD1) treatment for 72 hours or 15 ng/mL nocodazole treatment for 48 hours. Scale bars: 10 μm , 3 μm (zoom in, right). **b** Live cell images of the fusion and fission events of centrosomal OFD1 in Tet-inducible EGFP-OFD1-expressing RPE1 cells treated with 0.1 ng/mL Doxycycline and DMSO or 120 μM CK-666 for 72 hours. **c** Representative the SUM Z projection images of OFD1 and PCM1 staining in Synchronized hTERT-RPE1 treated with indicated siRNA, 120 μM CK-689, 120 μM CK-666 or 100 nM Cyto D for 72h (Left Panel). Quantitative data of OFD1 intensity for the staining (Right Panel), **** $P < 0.0001$, two-tailed unpaired student's *t*-test.

The proliferation assays with Ki67 must be performed on synchronized cells to allow for similar conditions.

Response: Thanks for the suggestion. The experiments were repeated with synchronized cells for at least three independent biological repeats, and the quantification was measured. The results were similar to that of non-synchronized cells (Supplementary Fig. 2e, Response Fig. 10).

Response Fig. 10. Actin debranching phenocopies OFD1 depletion in synchronized cells. a Representative images of Ki67 and ARL13B staining in Synchronized hTERT-RPE1 treated with indicated siRNA, 120 μM CK-689, 120 μM CK-666 or 100 nM Cyto D for 72h. Cilia are marked by arrowheads (Left Panel). Quantitative data for the staining (Right Panel). Data shown represent mean ± SD percentage of cells from triplicate samples. **** $P < 0.0001$, two-tailed unpaired student's t -test.

The HD-high density cultures are difficult to interpret since siOFD1 cells do not proliferate (Fig.3). How were these cultures produced in the first place?

Response: Thanks for the question. To produce the siOFD1 cells in high density, we passaged more normal cells to the dishes to form high-density cultures and carried out OFD1 knockdown at the same time.

The RNAseq data in Fig3c needs to show control cells, biological replicas, and statistics.

Response: Thanks for the suggestion. As the reviewer suggested, we reorganized the RNAseq data and presented the data with controls (Fig. 3c, Response Fig. 11).

Response Fig. 11. RNASeq analysis upon OFD1 depletion. a Analysis of RNA expression of wild-type RPE1 cells and OFD1-depleted RPE1 cells by RNAi. The transcription of genes related to G₀ and G₁ were shown in the chart. Each column is a biological replicate.

The OFT20 and CEP164-/cell-based studies have not been performed on synchronized cells. The defects of CEP164 are well described in cancer cells, and depletion causes hyper-proliferation. In this regard, the current data contradict some published data.

Response: Thanks for the suggestion. We performed the experiments in synchronized cells and found that the percentage of cells with cell cycle arrest and cilia formation remained comparable between synchronized or non-synchronized samples (Response Fig. 10). Besides, with OFD1 depletion, like serum starvation, most of the cells entered into quiescence, further disrupted cilia by knockout *IFT20* or *CEP164* did not rescue OFD1 depletion-induced quiescence. Consistent with the publication, our data also showed that cells with CEP164 deletion grew slightly faster (Fig. 5a, b). Please see the Y-axis of the two panels. We also merged the data in Response Fig. 12.

Response Fig. 12. OFD1 depletion on cell proliferation. a Proliferation curves of RPE1 cells expressing SV40 T Antigen and *CEP164*^{-/-} RPE1/TAg cells. Data shown represent mean \pm SD.

The p53 KO and Tag data in RPE cells lack the controls for the basal levels of proliferation and cell death. The conclusion that Tag in RPE cells somehow allows bypassing G₀ in OFD1 KO lacks experimental evidence.

Response: Thanks for the comments. The basal control state of p53 KO cells was added in Supplementary Fig. 3h, i. The proliferation of RPE1-SV40-TAG was described and quantified in Fig. 5a, d, e, f, and h. The data shown RPE1-SV40-TAG bypassing quiescence were presented in Supplementary Fig. 4i, j by Ki67 staining. We also observed that RPE1-SV40-TAG cells could bypass cell cycle arrest into S- and M-phase under OFD1 depletion by live cell imaging experiments in Fig. 5f. To address the reviewer's concern, we performed the cell cycle analysis experiments by flow cytometry. It showed that RPE1-SV40-TAG cells could bypass G₀ arrest under OFD1 depletion (Response Fig. 13h).

Response Fig. 13. Tag transformed cells bypass OFD1 depletion induced quiescence independent on p53. **a** Representative images of γ -tubulin and Ki67 staining of WT and OFD1 knockdown RPE1 cells with indicated genotype background. Cells were subjected to DMSO or 125 nM centrinone treatment for 72 hours. Cilia are marked by arrowheads. **b** Quantification of cells with positive γ -tubulin (centrosome) and Ki67 staining in **a**. Data shown represent mean \pm SD percentage of cells from triplicate samples. *** $P < 0.001$, two-tailed unpaired student's *t*-test. **c** Proliferation curves of RPE1 cells expressing SV40 T Antigen with or without knockdown of OFD1. Data shown represent mean \pm SD. **d** Mitotic events of WT

(Left Panel) and OFD1-depleted (Right Panel) RPE1/TAg cells were monitored by phase-contrast time-lapse microscopy. Yellow arrows and arrowheads point at control cells undergoing normal M-phase. White arrows indicate OFD1-depleted cells undergoing mitotic cell death. Orange and red arrows point at OFD1-depleted cells with the mitotic failure. Nuclei were labelled with Hoechst. **e** Quantitation of the duration of M phase in cells transfected with indicated siRNAs. Data shown represent mean \pm SD. **f** Mitotic events of WT (Upper Panel) and OFD1-depleted (Lower Panel) RPE1/TAg cells were monitored by fluorescence time-lapse microscopy. RPF-LifeAct (red) and GFP-tubulin (green) were used to visualize F-actin and microtubules, respectively. Arrowheads point at the position of actomyosin rings. **g** Quantitation of the indicated mitotic events of cells transfected with indicated siRNAs. Data shown represent mean \pm SD. Data shown represent mean \pm SD, *** $P < 0.001$, two-tailed unpaired student's *t*-test. **h** DNA content analysis by flowcytometry for RPE1/TAg cells transfected with control or OFD1 siRNA, treated with 100 ng/mL nocodazole or untreated. **i** Expression of SV40 T Antigen abolishes OFD1 loss-induced cell cycle arrest but not cilia formation. Representative immunofluorescence images of RPE1 cells stained for ARL13B and Ki67. Cilia are marked by arrowheads. Data shown represent mean \pm SD. **j** Percentage of cells positive for ARL13B or Ki67 are from triplicate samples. *** $P < 0.001$, two-tailed unpaired student's *t*-test.

To clarify the effects of OFD1 on cilia vs. actin dynamics around centrosome, rescue experiments with OFD1 point mutants at the Arp2 binding site in CK-666 treated cells are needed. Current data does not uncouple the generic effect of CK-666 on actin dynamics in cells from the centrosome/actin interface and thus might be indirect effects.

Response: Thanks for the comment. We generated siRNA-resistant doxycycline-inducible OFD1-WT and OFD1-W1012A mutant RPE1 cells. Upon depletion of endogenous OFD1, the expression of OFD1-WT at appropriate levels rescued OFD1 depletion-induced quiescence and ciliogenesis, while the expression of OFD1-W1012A mutant failed to do so (Fig. 4a, b and Supplementary Fig. 4f-h, Response Fig. 5).

We tested if the interaction between OFD1 and Arp2/3 complex is required for centrosomal actin branching *in vivo*. We evaluated the rescue functions of siRNA-resistant EGFP-OFD1 and EGFP-OFD1-W1012A mutant by comparing the centrosomal F-actin under OFD1 depletion in HeLa cells. The centrosomal F-actin clouds were dramatically reduced in cells with expression of EGFP-OFD1-W1012A mutant compared to that of EGFP-OFD1-WT cells (Supplementary Fig. 1j, Response Fig. 2).

Response Fig. 5. OFD1 regulates cell cycle and ciliogenesis via controlling centrosomal actin branching. **a** Representative images of ARL13B and Ki67 staining in Tet-inducible EGFP-OFD1-expressing RPE1 cells with indicated titration of Doxycycline and indicated siRNAs for 72 hours. **b** Quantitation of cells positive for ARL13B or Ki67 staining in **a**. Data shown represent mean \pm SD percentage of cells from triplicate samples. **** $P < 0.0001$, two-tailed unpaired student's t -test. **c** Representative images of ARL13B and Ki67 staining in Tet-inducible EGFP-OFD1-W1012A-expressing RPE1 cells with indicated titration of Doxycycline and indicated siRNAs for 72 hours. Cilia are marked by arrowheads. **d** Quantitation of cells positive for ARL13B or Ki67 staining in **c**. Data shown represent mean \pm SD percentage of cells from triplicate samples. **** $P < 0.0001$, two-tailed unpaired student's t -test. **e** Immunoblot analysis for GFP and α -tubulin of Tet-inducible EGFP-OFD1 or EGFP-OFD1-W1012A expressing RPE1 cells with indicated titration of Doxycycline and indicated siRNAs. The numbers under the gel lanes represent the ratio of EGFP-OFD1 band intensity to α -tubulin band intensity, which were normalized relative to the line 5 sample.

The data in Fig4f-g contradict the original premise as cells with depletion of RB1\RB11\RBL2 have 80% ciliated cells entering the cell cycle and Ki67positive (S phase). The RB pathway also directly influences actin dynamics via ILK and mDia1.

Response: Thanks for the comments. Depletion of RB1\RBL1\RBL2 rescued OFD1 depletion induced quiescence, however these cells still had cilia. Here the data emphasized that OFD1 depletion caused cell cycle arrest and cilia formation in RPE1 cells. Though the quiescence was dependent on RB pathway, RB inactivation was not sufficient to restore cilia disassembly. It was well documented that most types of cells still preserve cilia during G₁, S, and G₂, but cilia are disassembled before mitosis to release the centrosomes for spindle pole formation. Many proteins are involved in the actin network regulation at various degrees. Though RB1 may regulate ILK and mDia1, our results showed the formation of cilia was not significantly changed with or without RB family (Fig. 4f, g, Response Fig. 14), it is possible that RB loss impose subtle influence on centrosomal actin network that directly impacts ciliogenesis.

Response Fig. 14. OFD1 depletion induced quiescence is dependent on RB pathway. **a** Representative immunofluorescence staining of ARL13B and Ki67 in RPE1 cells with indicated treatments. Cilia are marked by arrowheads. **b** Quantitation of ARL13B or Ki67-positive cells in **a**. Data shown represent mean \pm SD, *** $P < 0.001$, two-tailed unpaired student's *t*-test. Three independent experiments have been performed for each part of this figure.

The drastic decrease in proliferation upon co-expression of hTERT and SV40 Tag was previously described and associated with significant cell death with higher basal levels of divisions and ki67 positivity. The authors called these cells transformed without evidence to support malignant transformation.

Response: Thanks for the comments. Expression of SV40-TAg was considered to be a widely used method to immortalize cells. Published studies showed that co-expression of hTERT and Tag induced very mild cell death, but the expression of TAg and hTERT largely promoted cell growth. The increased proliferation of the cells co-expressing TAg and hTERT was also observed in our studies. Consistent with the publication (Kobayashi, Front Cell Dev Biol. 2020, PMID: 33251215), cell death will be compensated by a high proliferation rate. To address this concern, we also determined the percentage of cell death in SV40-TAg transformed cells. As shown in the Response Fig. 15, few cell death was observed.

Response Fig. 15. FACS analysis upon TAG transformation. **a** Detection of Cell viability RPE1 cells and RPE1/TAG cells by Annexin V-PI staining assay (Upper Panel). Cell morphology was examined by a bright-field microscope (Lower Panel).

The role of OFD1 in cancer is controversial, and the data shown in Extended Fig.6c-e requires more depth on how the panels were generated with specific database links and statistics. The analysis of OFD1 expression in protein atlas <https://www.proteinatlas.org/ENSG00000046651> OFD1/pathology or <https://kmplot.com/analysis/> shows that for some cancers up-regulation of OFD1 is a favorable prognostic marker and few cancers have a significant difference in OS (overall survival) based OFD1 expression.

Response: Thanks for the comments. In the **Supplementary Fig. 6c** (Response Fig. 16), the public normalized gene expression data based on fragments per kilobase of exon model per million reads mapped (FPKM) of tumor samples, and adjacent normal samples were obtained from The Cancer Genome Atlas (TCGA) data portal (<http://gdac.broadinstitute.org/>). Then we performed Wilcoxon signed-rank test to assess the different expression between tumor tissues and adjacent normal tissues.

Response Fig. 16. OFD1 is overexpressed in a broad spectrum of tumors. a Data summary of OFD1 mRNA levels normalized from fragments per kilobase of exon model per million reads mapped (FPKM) of tumor samples and adjacent normal samples obtained from The Cancer Genome Atlas (TCGA). Wilcoxon signed-rank test assesses the different expression between tumor tissues and adjacent normal tissues.

We perform analyses for patients from The Cancer Genome Atlas (TCGA) and reveal divergent patterns for OFD1 expression features across multiple cancer types.

We also used the following common website for analysis, <http://ualcan.path.uab.edu/cgi-bin/TCGAExResultNew2.pl?genenam=OFD1&ctype=CHOL> (Response Fig. 17).

Response Fig. 17. OFD1 is overexpressed in a broad spectrum of tumors. a b The public OFD1 mRNA levels data from The Cancer Genome Atlas (TCGA) were measured in TPMs (Transcripts per Millions) from the common website, UALCAN, and revealed divergent patterns for OFD1 expression features across multiple cancer types.

The data did not mean that OFD1 were overexpressed in all cancer type, neither did we claimed that OFD1 expression levels correlate with survival. Interestingly, at least in some cancer types, such as Cholangiocarcinoma and lung cancer, compared to normal tissues, OFD1 was aberrantly overexpressed in tumor tissues. Similarly, the expression levels of well-known oncogene *Kras* or *Src*, were not always consistent in different tumor types, and the high expression levels of these proteins positively or negatively correlate with survival in a context dependent manner (Response Fig. 18).

a

b

c

d

Response Fig. 18. KRAS and SRC expression in multiple tumors. a The analysis of KRAS expression in *ProteinAtlas*. **b** The analysis of KRAS expression in *KMPlot*. **c** The analysis of SRC expression in *ProteinAtlas*. **d** The analysis of SRC expression in *KMPlot*.

The overall study is comprehensive and provided evidence to support the notion that centrosome-associated OFD1 is essential for actin accumulation and control of ciliary dynamics. However, its role in cell proliferation and mitosis seems more complicated and potentially actin independent.

Response: Thanks for the comments. To investigate the role of OFD1 regulated actin dynamics in cell cycle control, we performed more experiments in the revised version. We showed that disruption of the interaction of OFD1 with Arp2/3 complex led to reversible cell cycle arrest in normal cells and caused irreversible mitotic cell death in cancer cells, the phenotypes of which were consistent with OFD1 depletion. These results indicated the specificity of OFD1-regulated actin dynamics in cell cycle control and support the conclusion that the role of OFD1 in cell proliferation and mitosis mainly relies on actin dynamics. Evidences from biochemical and cell biology studies support the conclusion: 1) Disruption of actin branching by genetic depletion of Arp2/3 components or chemical inhibition, phenocopied OFD1 silencing induced quiescence with cilia formation in normal RPE1 cells, which is distinct from centrosome surveillance checkpoint, and was RB dependent but cilia independent. 2) Complement of OFD1-WT but not the ARP2 binding deficient mutant OFD1-W1012A, rescued OFD1 depletion induced cell cycle arrest and ciliogenesis. 3) OFD1-WT-peptide but not OFD1-W1012A mutant peptide, competed the OFD1-Arp2/3 complex binding, inhibited VCA-OFD1-Arp2/3 complex mediated actin branching, and phenocopied OFD1 depletion induced quiescence with ciliogenesis.

Reviewer #3 (Remarks to the Author):

In this manuscript, Cao and his colleagues describe a novel role of OFD1 in cell cycle control through regulating actin branching at the centrosome region, which affects several cellular events including ciliogenesis, cytokinesis and cell cycle restriction point. They first identified OFD1 as a novel Class II actin nucleation promoting factor which promotes actin branching around the centrosome. They further confirmed that its NPF function is important for interphase ciliogenesis and mitotic spindle assembly. Then they proved that OFD1 inactivation through siRNA or shRNA led to cell cycle arrest at G0 and ciliogenesis through the RB-dependent pathway. Intriguingly, they found that while OFD1 knockdown led to cell cycle arrest in normal cells, its inhibition greatly suppressed cancer cell proliferation and tumor growth. These results shed light on centrosome-actin, actin-ciliogenesis, actin-mitosis relationships and cancer therapies. Overall, this is a well-designed project with sufficient evidence to support the conclusions and should be published.

Response: Thank the reviewer for the appreciation of our work.

Minors and suggestions

1. Fig. 4f-4i showed that co-KD of RB1/L1/L2 could rescue cell cycle arrest caused by siOFD1, but did not cause cilia disassembly, suggesting OFD1 may have a direct role in ciliogenesis and cilia disassembly. Are the cilia length the same when co-KD RB proteins compare to OFD1 KD alone?

Response: Thanks for the suggestion. We measured the length of cilia that formed in the OFD1 KD and OFD1/RBs KD cells, and found that no obvious difference of ciliary length (Response Fig. 19).

Response Fig. 19. RBs KD has minor effect on ciliary length. a Quantification of ciliary length. Representative images of OFD1 and ARL13B staining in RPE1 cells treated with indicated siRNA (Left Panel). Quantitative data for the staining (Right Panel).

2. Actin filament destabilization by CK-666 or cytoD caused centriolar satellite proteins relocation. Does the actin network function in centriolar satellite maintenance? This need to be further discussed.

Response: Thanks for the professional suggestion. In the revised manuscript, we discussed this possibility in the section of Discussion (Line 445).

3. Line 324, CEP164^{-/-} cells also showed inhibited proliferation should add “when OFD1 is depleted” to be more accurate.

Response: Thank the reviewer for pointing out this. We revised the description as the reviewer suggested (Line 374).

4. OFD1 KD affects actin branching around the centrosome and Arp2/3 inhibition by CK-666 treatment inhibits actin branching more broadly. Did the authors directly compare the ability of OFD1 KD and CK-666 in inducing ciliation in the presence of serum?

Response: Thanks for the question. We compared the ability of OFD1 KD, CK-666, and Cyto D (Response Fig. 10).

Response Fig. 10. Actin debranching phenocopies OFD1 depletion in synchronized cells. a Representative images of Ki67 and ARL13B staining in Synchronized hTERT-RPE1 treated with indicated siRNA, 120 μM CK-689, 120 μM CK-666 or 100nM Cyto D for 72h. Cilia are marked by arrowheads (Left Panel). Quantitative data for the staining (Right Panel). Data shown represent mean ± SD percentage of cells from triplicate samples. **** $P < 0.0001$, two-tailed unpaired student’s *t*-test.

5. Images in Fig.2a are small. Showing zoomed crop regions like Fig.2c could help to illustrate the localization changes of OFD1.

Response: Thanks for the suggestion. We included the zoomed pattern to illustrate the changes of OFD1 localizations (Fig. 2a, Response Fig. 20).

Response Fig. 20. OFD1 is dynamically regulated during actin debranching. **a** hTERT-RPE1 cells were co-stained with antibodies against OFD1 and PCM1. Cells were subjected to DMSO, serum starvation, 120 μ M CK-689, 120 μ M CK-666, 100 nM Cyto D, 20 μ M SMIFH2 treatment for 96 hours, si-Control (control siRNA), si-OFD1 (siRNA targeted OFD1) treatment for 72 hours or 15 ng/mL nocodazole treatment for 48 hours. Scale bars: 10 μ m, 3 μ m (zoom in, right).

6. In Tet-on/off systems, Dox concentrations are typically used at 100 nm to 1 μ m. Here the authors described a Dox concentration as low as 0.01 nm. I wonder if this is a typo.

Response: Thanks for the comments. We induced the OFD1 expression by titrating Dox to 0.01 ng/ml, 0.1 ng/ml, or 1 ng/ml. We confirmed that these were not spelling mistakes. The Tet-on/off system we used in our study is very sensitive to Dox concentrations, since in this system, we incorporated the target gene to a transcriptional active region as a single copy, normally the gene expression is blocked by Tet repressor, and the expression of the target gene could be induced by an adjustable amount of Dox to a very low level depends on gene expression. We have described this system is in several publications (Sun, PNAS, 2008, PMID:19050071; Chen, Mol Cell. 2012, PMID:22342342; Tang, Nature, 2013, PMID:24089205).

Thank all the reviewers again for the constructive comments and valuable suggestions to improve the quality of our manuscript.

REVIEWERS' COMMENTS

Reviewer #1 (Remarks to the Author):

The authors have addressed my concerns.

Reviewer #2 (Remarks to the Author):

The manuscript "An Actin Branching Network Surveillance System Controls the Restriction Point, Cytokinesis and Primary Ciliogenesis" by Cao M is a comprehensive study of OFD1 function and its effects on cilia biology and cell cycle. The results of the study are significant and novel. The conclusions were supported by the solid research plan and sound methodology. The authors had completed rigorous revisions and improved on the depth, rigor and scientific scope of the study. There are few minor points to remedy before it can be accepted. Specifically:

1. The amount of OFD1 at the centrosome/PCM seems a minor population and nearly undetectable with endogenous antibody. In Figure 1e the colocalization with ARP2 visually undetectable (in green channel) there is no detectable signal in the same frame as in OFD1. Provide better resolution for ARP2/OFD1 imaging, including 3D XYZ projection to allow for dimensional resolution of ARP2 and OFD1 pools.
2. Similar comments for Figure 1f. The high background of actin does not allow to visualize the colocalization. The 3d XYZ images and improved fixation procedure might allow for the better resolution. the concern is also that the imaging depth of siCon and siOFD1 might be different and thus affecting the perception of actin branching.
3. Please include 3D XYZ projections, to allow for side view of cell depth in images Supplementary 1g and j.
4. Please discuss why the OFD1 at the centrosome changes within the same cell line (RPE) depending on the co-staining protein. Compare images Fig. 1e and Fig.2a.
5. High resolution images of cilia should be added to panels in Fig.2j and Fig.4a and 4f similar to Supplementary 4c to allow for visualization of the phenotype.

Reviewer #3 (Remarks to the Author):

The authors have satisfactorily addressed all my comments for this paper and answered my technical questions. The paper has been significantly improved after revision. I recommend the manuscript for publication.

Reviewer #1 (Remarks to the Author):

The authors have addressed my concerns.

Response: We thank review #1 for great suggestion and help us to improve the scientific logic and data quality of our manuscript.

Reviewer #2 (Remarks to the Author):

The manuscript "An Actin Branching Network Surveillance System Controls the Restriction Point, Cytokinesis and Primary Ciliogenesis" by Cao M is a comprehensive study of OFD1 function and its effects on cilia biology and cell cycle. The results of the study are significant and novel. The conclusions were supported by the solid research plan and sound methodology. The authors had completed rigorous revisions and improved on the depth, rigor and scientific scope of the study.

Response: We would like to thank the reviewer for careful and thorough reading of this manuscript and for the thoughtful comments and constructive suggestions, which help to improve the quality of this manuscript. Our response follows.

There are few minor points to remedy before it can be accepted. Specifically:

1. The amount of OFD1 at the centrosome/PCM seems a minor population and nearly undetectable with endogenous antibody. In Figure 1e the colocalization with ARP2 visually undetectable (in green channel) there is no detectable signal in the same frame as in OFD1. Provide better resolution for ARP2/OFD1 imaging, including 3D XYZ projection to allow for dimensional resolution of ARP2 and OFD1 pools.

Response: Great suggestions. The 3D XYZ projection have been updated as suggested.

2. Similar comments for Figure 1f. The high background of actin does not allow to visualize the colocalization. The 3d XYZ images and improved fixation procedure might allow for the better resolution. the concern is also that the imaging depth of siCon and siOFD1 might be different and thus affecting the perception of actin branching.

Response: Great suggestions. The 3D XYZ projection have been updated as suggested.

3. Please include 3D XYZ projections, to allow for side view of cell depth in images Supplementary 1g and j.

Response: Great suggestions. The 3D XYZ projection have been updated as suggested.

4. Please discuss why the OFD1 at the centrosome changes within the same cell line (RPE) depending on the co-staining protein. Compare images Fig. 1e and Fig.2a.

Response Figure 1. Enlarged region of OFD1 staining.

Response: The difference between the two images may be resulted from the resolution of the PDF file. In the revised manuscript, we replaced all the Figures with AI files. The localization of OFD1 remained consistently at the centrioles and the centriolar satellite in the enlarged images (Response Figure 1). It is also possible that the fixation condition may slightly affect the background. Using an improved fixation method for actin staining, 37°C for 10 min with fresh 4% paraformaldehyde (PFA) in the cytoskeleton preserving buffer (PEM) (80 mM PIPES pH 6.8, 5 mM EGTA, 2 mM MgCl₂) (PMID: 27378434; PMID: 31024536) was applied in Fig.1e. In Fig. 2a, methanol was applied for observing centriolar satellite proteins (PMID: 20230748, PMID: 33934390).

5. High resolution images of cilia should be added to panels in Fig.2j and Fig.4a and 4f similar to Supplementary 4c to allow for visualization of the phenotype.

Response Figure 2. Representative enlarged images of cilia.

Response: We thank the reviewer for the suggestion. The previous images in PFD files have been compressed, so the cilia may be not that clear. For example, cilia in Fig.2j can be clearly observed in the images (Response Figure 2). In the revised manuscript, we enlarged the images if there is enough space and replaced all the Figures with AI files. Meanwhile, cilia were marked with arrow heads.

Reviewer #3 (Remarks to the Author):

The authors have satisfactorily addressed all my comments for this paper and answered my technical questions. The paper has been significantly improved after revision. I recommend the manuscript for publication.

Response: We appreciate the Reviewer #3 for reviewing our manuscript. We also greatly appreciate the reviewers for the constructive comments and suggestions.